# Time-dependent catalytic activity in aging condensates

Wei Kang [1,2,6] ✉, Zhiyue Wu[1,6], Xinzhi Huang [3,6], Hongbin Qi[1], Jiaxuan Wu[4], Jiahui Wang[1], Jing Li [5], Sijin Wu[4], Byung-Ho Kang [5], Bo Li [3] ✉, Juncai Ma [5] ✉ & Chuang Xue [1,2] ✉

Biomolecular condensates are dynamic cellular compartments that concentrate proteins and enzymes to regulate biochemical reactions in time and space. While these condensates can enhance enzyme activity, how this function changes as condensates age remains poorly understood. Here, we design synthetic catalytic condensates that selectively recruit enzymes to investigate this temporal evolution. We show that catalytic condensates exhibit time-dependent activity: they initially accelerate enzymatic reactions but gradually lose efficiency due to the transition from liquid-like to solid-like states. This aging process, characterized by protein aggregation and loss of selective barriers, impairs enzyme function both in vitro and living cells. We further demonstrate that small molecules which influence aging dynamics can modulate catalytic efficiency of condensates. Our findings show that condensate aging as a key regulator of enzymatic activity and provide crucial insights for designing functional synthetic condensates.

The organization of proteins into discrete self-assemblies is fundamental for cells to achieve spatial regulation of cellular components and orchestrate biochemical reactions. For example, the formation of multi-enzyme complexes, such as fatty acid synthases and polyketide synthases, is crucial for enhancing and regulating enzymatic flux[1,2]. Recent studies have identified ubiquitous protein assemblies known as biomolecular condensates (or membraneless organelles), which form through liquid-liquid phase separation (LLPS) of proteins and/or nucleic acids driven by weak and transit interactions, including nucleoli[3], stress granules[4] and P granules[5]. Cells utilize biomolecular condensates to regulate and augment the functions of the encapsulated enzymes[6–8]. For example, condensates containing ribulose bisphosphate carboxylase/oxygenase (Rubisco) enhance overall efficiency and mitigate inherent selectivity challenges during carbon fixation[9,10]. Inspired by naturally occurring condensates, researchers have engineered synthetic condensates by exploiting the intrinsically disordered regions (IDRs) to increase product yield and enable programmable control over cellular behavior[11–16]. The ability of biomolecular condensates to participate in and influence diverse fundamental and synthetic biological functions arises from unique properties—distinct from the cytosolic environment—conferred by collections of assembled biomolecules that individual molecules lack[7,17]. The mechanisms through which specific properties influence the functions of phase-separated biochemical processes are becoming increasingly characterized[18–22].

However, current studies predominantly based on the premise that biomolecular condensates remain in a liquid-like equilibrium state[17,23]. Nonetheless, condensate aging, wherein biomolecular condensates formed via LLPS transition from an initially dynamic, liquid-like state to less dynamic or even solid-like states over time, has been

[1]MOE Key Laboratory of Bio-Intelligent Manufacturing, State Key Laboratory of Fine Chemicals, Frontiers Science Centre for Smart Materials Oriented Chemical Engineering, School of Bioengineering, Dalian University of Technology, Dalian, China. [2]Ningbo Institute of Dalian University of Technology, Ningbo, China. [3]Department of Mechanical Engineering, Kennesaw State University, Marietta, GA, USA. [4]Jiangsu Province Higher Education Key Laboratory of Cell Therapy Nanoformulation, Wisdom Lake Academy of Pharmacy, Xi'an Jiaotong-Liverpool University, Suzhou, China. [5]School of Life Sciences, Centre for Cell & Developmental Biology and State Key Laboratory of Agrobiotechnology, The Chinese University of Hong Kong, Shatin, New Territories, Hong Kong, China. [6]These authors contributed equally: Wei Kang, Zhiyue Wu, Xinzhi Huang. ✉e-mail: kangwei@dlut.edu.cn; bli10@kennesaw.edu; juncaima@cuhk.edu.hk; xue.1@dlut.edu.cn

frequently observed in cases involving FUS[24], tau[25], a-synuclein[26], and TDP43[27]. Cells leverage condensate aging to impart a broad spectrum of properties—from viscous liquids to elastic solids—enabling the versatile biological functions of these condensates in living systems[28–33]. For example, a recent study has shown that the solid-state of an RNA-protein condensate, namely *oskar* granules, is essential for RNA localization and translation during *Drosophila* embryonic development[34]. Conversely, the liquidity of viral condensates is required for virus to accomplish essential life cycle functions[30]. Furthermore, dysregulation of condensate aging is intimately linked to neurodegenerative disease[35–37].

Biomolecular condensates may exhibit varying biophysical properties adapted to specific functions, underscoring the importance of these properties in influencing biological roles. However, it remains unclear how these impacts change over time due to the complex nature of the aging process. Although several studies have tried to elucidate the mechanisms behind condensate aging[28,38–41], a systematic understanding of its biological implications is still lacking. While some reports suggest that condensate aging is related to a glass transition without significant structural changes[42], others indicate that condensate aging may lead to gelation alongside the accumulation of fibrillar structures[24,26]. A further challenge in studying the impact of condensate aging lies in the limited methods available for direct evaluation within living cells, often leading researchers to rely on reconstituted non-equilibrium condensates in vitro. However, the aging behavior of intracellular condensates remains largely untapped.

In this work, we study how condensate aging affects biological functions of aging condensates both in vitro and in living cells by engineering a synthetic condensate capable of catalyzing enzymatic reactions. The phase-forming scaffolds of the synthetic condensates comprise an IDR from fibrillarin and an interaction domain (RIAD)[43], enabling them to undergo phase separation and recruit target enzymes containing the cognate peptide RIDD. We simultaneously tracked changes in the enzymatic activity and the biophysical properties of these catalytic condensates over time. In contrast to many previous studies reporting enhanced reaction rates with synthetic condensates[12,16], we observe a time-dependent enzyme activity characterized by transient enhancement followed by a gradual decrease in reaction rate. Through a combination of biophysical characterization, including particle tracking microrheology (PTM) and fluorescence recovery after photobleaching (FRAP), we reveal that the initially liquid-like condensate transitions into a Maxwell fluid over time, and finally to solid-like state. Transmission electron microscopy (TEM) and electron tomography (ET) revealed the formation of protein aggregates within aged condensates both in vitro and in living cells. In addition, we found that the aged condensates lose the phase-separated barrier against surrounding cytosolic contents. We propose that the formation of protein aggregates and the loss of selective permeability contribute to the malfunction of the aged condensates. Furthermore, we demonstrate a partial rescue of impaired enzymatic activity in aged condensates through the application of small molecules that delay the aging process. Our findings suggest a potential role of condensate aging in regulating chemical reactions and provide important information for the rational design of synthetic condensates aimed at augmenting cellular functions.

## Results

### Prolonged LLPS compromises the enhanced activity of the synthetic catalytic condensate

We utilized fibrillarin, a key nucleolar protein involved in forming dense fibrillar components, to construct synthetic catalytic condensates. Recent studies have shown that full-length fibrillarin can undergo phase separation into a protein condensate[3]. In silico analysis revealed that the N-terminal region of fibrillarin is an intrinsically disordered regions (IDR), a feature crucial for its phase-separating properties (Supplementary Fig. 1). To simplify expression and mitigate potential challenges associated with full-length fibrillarin, we selected the IDR of fibrillarin, henceforth referred to as FIB1, as the phase-forming scaffold for our synthetic condensates. To selectively recruit and enrich target enzymes within the condensates, we fused our previously developed complementary interaction peptides, RIAD and RIDD[43], to FIB1 and enzyme, respectively. This specific RIAD-RIDD interaction was designed to facilitate the targeted recruitment of enzymes into synthetic condensates. For visualization, we inserted a green fluorescent protein (GFP) tag between FIB1 and RIAD, yielding the phase-forming construct FIB1-GFP-RIAD (Fig. 1a). This construct was then expressed and purified to homogeneity using an *E. coli* expression system (Supplementary Fig. 2). We first assessed the phase separation of FIB1-GFP-RIAD in vitro using confocal microscopy. The phase diagram revealed that the protein condensate formation was favored at lower NaCl concentrations and higher protein concentrations, while higher NaCl concentrations and lower protein concentrations inhibited condensate formation (Supplementary Fig. 3). For subsequent experiments, we used a protein concentration of 6 μM and a condensate formation buffer (150 mM NaCl, 20 mM Tris, 0.5 mM 4-(2-aminoethyl)-benzenesulfonyl fluoride hydrochloride (AEBSF), 0.1 mM dithiothreitol (DTT), 0.1% (v/v) Triton X-100, and 10% (v/v) glycerol, pH 7.4). Under these conditions, FIB1-GFP-RIAD formed spherical condensates approximately 5 μm in diameter (Supplementary Fig. 4).

Next, we tested the feasibility of enzyme recruitment by incorporating RIDD-mCherry as a model client protein (Fig. 1b). The partitioning of client proteins into the condensates was quantified using an enrichment index, defined as the ratio of the fluorescence intensity of client proteins inside versus outside the condensates. Confocal imaging showed that mCherry without the RIDD tag exhibited nearly identical fluorescent intensities inside and outside the condensates, yielding an enrichment index of approximately 1.0 (Fig. 1b and c). In contrast, fusion of the RIDD tag to mCherry resulted in a significant enrichment, with an index of 10.7 (Fig. 1b and c). Control experiments demonstrated that fusion of either full-length FIB1 or a truncated version, FIB1$_{0.5}$, to mCherry resulted in minimal enrichment within the condensates (Fig. 1b and c). These results confirm the effectiveness of the modular RIAD-RIDD system for selective enzyme recruitment into synthetic condensates.

To examine the impact of condensate aging on enzymatic function, we monitored the rate of a model enzymatic reaction catalyzed by MenH, a key enzyme in menaquinone biosynthesis (Fig. 1d). MenH catalyzes the conversion of 2-succinyl-5-enopyruvyl-6-hydroxy-3-cyclohexadiene-1-carboxylate (SEPHCHC) to 2-succinyl-6-hydroxy-2,4-cyclohexadiene-1carboxylate (SHCHC) through pyruvate elimination (Fig. 1d), with SHCHC formation detectable by absorbance at 293 nm[44]. We compared reaction rates of catalytic condensates at three aging stages: immediately after condensate formation (h0), 4 h post-formation (h4), and 8 h post-formation (h8). We confirmed the enrichment of MenH-RIDD inside the condensates (Supplementary Fig. 5). For each age, reaction solutions incubated for the same duration and containing an equivalent amount of enzyme, served as the control.

We observed a gradual decrease in reaction rate within the catalytic condensates from h0 to h8 (Fig. 1e and f). At h0, the reaction rate was 18.7 μM/min, a 1.5-fold increase over the control, indicating enhanced enzymatic activity (Fig. 1e and f). This enhancement aligns with previous studies demonstrating an increased reaction rate for MenH in synthetic condensates[45]. However, by h4, the reaction rate in the condensates decreased to 13.2 μM/min, comparable to the control (Fig. 1e and f). At h8, prolonged LLPS resulted in a reaction rate reduced by a factor of 6.2 compared to the control (Fig. 1e and f). To

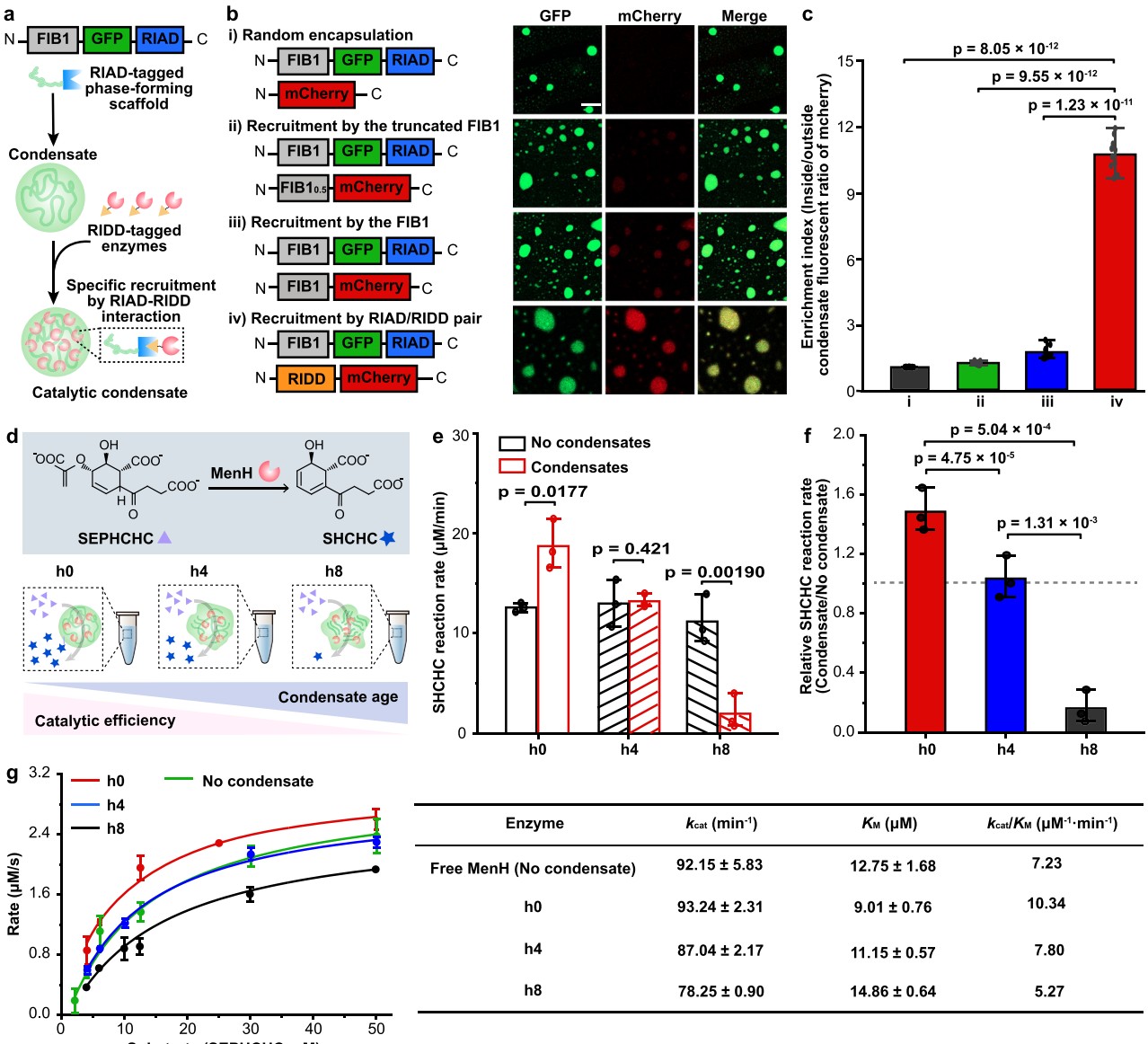

**Fig. 1 | Enzymatic reactions within condensates exhibit a time-dependent reaction rate. a** Schematic for the design of the catalytic condensate. The phase-forming scaffold consists of the intrinsically disordered region (IDR) of fibrillarin (FIB1) responsible for liquid-liquid phase separation, a fluorescent tag (GFP) responsible for visualization, and an interaction domain (RIAD) responsible for the recruitment of cargo enzymes containing the cognate RIDD domain. **b** Comparison of cargo protein enrichment within condensates between different strategies. The schematic illustrates cargo protein recruitment strategies (left), including i) random encapsulation, ii and iii) attaching the cargo protein with the identical IDR region (FIB1) or truncated IDR (FIB1$_{0.5}$), and iv) attaching cargo protein and phase-forming scaffold with cognate interaction motifs RIDD and RIAD, respectively. Representative images of mixtures of cargo proteins and phase-forming scaffold showing the strongest enrichment of cargo protein is achieved by using strategy iv (right). Scale bar, 10 μm. **c** Quantification of data as in (**b**), showing the enrichment indexes for different strategies, defined as the fluorescence intensity of mCherry inside versus outside the condensates. Data are represented as mean ± SD from 10 fields ($n = 10$) from three independent experiments (≥ 100 condensates). **d** Schematic diagram depicting the MenH catalyzed reaction. The enzymatic activity of the catalytic condensates with different ages (hn, n hours after condensate formation) was determined, respectively. **e** Comparison of SHCHC reaction rates in condensates at varying ages. For each age, the enzyme system was incubated for equivalent durations and contained an equal amount of enzymes serving as a control. **f** Comparison of relative reaction rates. The normalized reaction rates were calculated as the ratio of reaction rates between the condensate and the corresponding control system. The gray dashed line serves as a visual indicator where a relative reaction rate of 1 suggests comparable reaction rates between the condensate and the control. **g** Comparison of enzyme kinetic between catalytic condensates at various ages, yielding $k_{cat}$ and $K_M$ reported in the right table. For **e**–**g**, data are represented as mean ± SD from three independent experiments ($n = 3$). Source data are provided as a Source Data file.

further investigate, we determined the initial reaction rates at varying substrate concentrations. Fitting the data to the Michaelis-Menten equation allowed us to calculate the enzyme kinetic parameters $K_M$ and $k_{cat}$. At h0, we noted a 29.3% lower $K_M$, with no significant change in $k_{cat}$, resulting in a 1.4-fold increase in catalytic efficiency compared to the free enzyme system (Fig. 1g). However, with prolonged incubation, the catalytic efficiency progressively decreased, with a 27.1% reduction in efficiency at h8 (Fig. 1g). These findings suggest that the decline in reaction rates over time is likely due to deteriorating enzyme kinetics. Our results demonstrate that while concentrating enzymes within condensates may initially enhance catalytic activity, prolonged LLPS impairs enzymatic function.

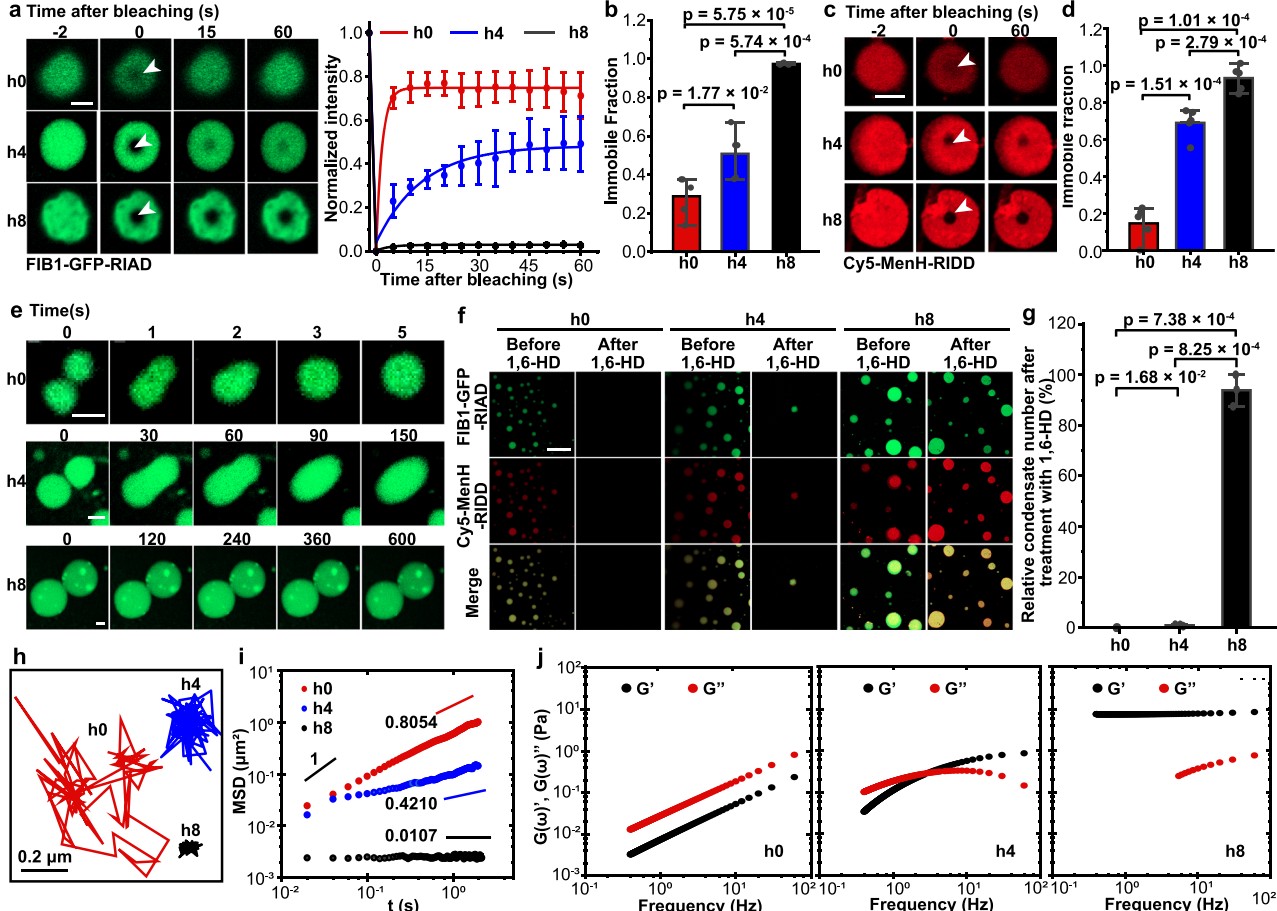

**Fig. 2 | Aging of catalytic condensate leads to a transition from a viscous liquid state to an elastic solid state via a Maxwell fluid phase. a** FRAP analysis of the phase-forming scaffold (FIB1-GFP-RIAD) in catalytic condensate at h0, h4 and h8, showing a progressive reduction in molecular mobility. Representative images of condensate before bleaching, during bleaching, and after bleaching are shown (left). FRAP recovery curves for FIB1-GFP-RIAD in h0, h4 and h8 condensates are presented (right). Bleached regions are indicated by white arrows. Scale bar, 2 μm. **b** Immobile fractions of FIB1-GFP-RIAD derived from **a**. **c** FRAP analysis of enzyme MenH-RIDD in h0, h4 and h8 catalytic condensates. Representative images of condensates obtained during FRAP experiments are shown. Bleached regions are indicated by white arrows. Scale bar, 2 μm. **d** Immobile fractions of the enzyme are presented. **e**, Confocal fluorescence images of catalytic condensate fusion. At h0, adjacent condensates coalesce within 3 s to form a larger single condensate, whereas 8-hour-aged condensates resist fusion. Scale bars, 2 μm. **f** Confocal images of catalytic condensates before and after 1,6-hexanediol (HD) treatment. Representative confocal images are shown. Scale bar, 20 μm. **g** Quantification of relative condensate number after treatment with 1,6-HD. Relative numbers are expressed as percentages compared to the condensate count before treatment. Data are presented as mean ± SD from three independent experiments ($n = 3$). **h** Representative probe particle trajectories in h0, h4 and h8 catalytic condensates in particle tracking microrheology (PTM). **i** Averaged MSDs versus lag time τ for probe particles in h0, h4 and h8 condensates. **j** Storage modulus G'(ω) and loss modulus G" (ω) as a function of frequency ω in h0, h4 and h8 condensates that are determined from PTM. For **a**–**d**, data are represented as mean ± SD from five independent experiments ($n = 5$). For **c** and **f**, a 1:9 (v/v) ratio of Cy5-labeled to unlabeled MenH-RIDD was used to visualize the enzymes. Catalytic condensates consist of 6 μM phase-forming scaffolds (FIB1-GFP-RIAD) and 1 μM of enzymes (MenH-RIDD) in condensate formation buffer. Source data are provided as a Source Data file.

## The liquid-like catalytic condensates transition from a viscous liquid state to an elastic solid state via a Maxwell fluid phase over time

To investigate the relationship between the time-dependent decline in reaction rates and potential changes in the biophysical properties of these non-equilibrium condensates, we concurrently monitored their biophysical characteristics. We began by assessing molecular mobilities within the condensates at various stages using fluorescence recovery after photobleaching (FRAP) analysis. At h0, we observed rapid replenishment of the bleached area with non-bleached molecules, resulting in a high recovery of fluorescent signal and a short halftime of recovery ($\tau_{1/2}$) of 0.5 s (Fig. 2a and Supplementary Fig. 6). This indicates a highly mobile microenvironment in the h0 condensates. However, both the recovery rate and the percentage of recovery diminished over time, reflecting a loss of dynamic characteristics within the condensates (Fig. 2a and Supplementary Fig. 6).

FRAP analysis of aged condensates (h8) revealed no observable fluorescence recovery, suggesting a transition to a nearly immobile state (Fig. 2a). The immobile fraction at h8 (97.4%) was approximately 1.9 and 3.4 times higher than those at h4 (50.8%) and h0 (28.8%), respectively (Fig. 2b). Based on $\tau_{1/2}$, we estimated that the apparent diffusion coefficients ($D_{app}$) decreased with prolonged LLPS, from 1.04 μm²/s (h0) to 0.06 μm²/s (h4) (Supplementary Fig. 6). Consistent with this, FRAP analysis of MenH-RIDD demonstrated a similar decrease in molecular mobility over time (Fig. 2c and d). The loss of dynamics in the non-equilibrium condensate was further corroborated by fusion experiments (Fig. 2e). Collectively, these findings demonstrate that condensate aging leads to a transition from a dynamic state to a dynamically arrested state.

To discern whether the observed decrease in molecular dynamics within the condensates resulted solely from increased viscosity or from rheological property change (i.e., transition from a viscous fluid

to an elastic solid), we first evaluated the state of matter of the condensates upon the addition of 10% (w/v) 1,6-hexanediol (1,6-HD), an aliphatic alcohol known to disrupt phase separation by weakening hydrophobic interactions[8]. We found that the addition of 1,6-HD led to the dissolution of freshly formed condensates, yet it failed to disrupt the h8 condensates, which retained both phase-forming scaffolds and enzymes (Fig. 2f and g), supporting the notion of a liquid-to-solid transition. We also probed the state of matter of the condensates through dilution experiments. Notably, the h0 condensates dissolved upon a 20-fold dilution (Supplementary Fig. 7), affirming their liquid state. Conversely, dilution did not affect the aged condensates at h8, indicating a more solid-like character (Supplementary Fig. 7). Together, these results demonstrate that the aging of condensates induces a transformation in their material properties, shifting from a liquid state to a solid state.

We further scrutinized the rheological property of aging condensates via particle tracking microrheology (PTM). In short, PTM is a passive microrheological technique that measures the rheological properties of materials by tracking the Brownian motion of embedded microscopic particles. Anomalous diffusion, characterized by deviations from the classical Brownian motion governing particle dispersion, is a hallmark of numerous complex soft-matter and biological systems[46,47]. The study of dynamic behaviors in these systems has been significantly advanced by PTM[48,49], which enables the investigation of local mechanical properties in both synthetic materials and live biological cells and tissues[50]. Notably, the frequency range probed by the particle tracking experiment is inherently limited by the experimental time scale and resolution. Therefore, the reported material properties apply only to the specific experimental time scale or frequency range used. In this study, we employed PTM to investigate the aging process by monitoring small fluorescent probe particles suspended within the condensates. The diffusive motion of multiple probe particles was tracked over time. Representative trajectories of probe particles in condensates at h0, h4 and h8 are illustrated in Fig. 2h. As the condensates undergo solidification, the motion of probe particles becomes progressively more restricted. Using PTM, the ensemble-averaged mean-squared displacements (MSD) of particles $\langle \Delta r^2(\tau) \rangle$ is determined as

$$\langle \Delta r^2(\tau) \rangle \equiv \langle (r(\tau) - r(0))^2 \rangle = 4 D_{app} \tau^{\alpha(\tau)} \tag{1}$$

Where $\langle \cdot \rangle$ represents the ensemble average over many particle trajectories, $\tau$ is the lag time, $\alpha(\tau)$ is the diffusive exponent, and $D_{app}$ is the apparent diffusion coefficients of probe particles in catalytic condensates. $D_{app}$ of h0 and h4 condensates are at the same order of magnitude that tested by FRAP (Supplementary Fig. 8). However, $D_{app}$ of h8 approaches infinity towards 0 µm²/s calculated by PTM, which aligns with the results that $D_{app}$ of h8 cannot be calculated by FRAP experiment (Supplementary Fig. 8). The dynamics of condensates tested by FRAP and PTM are mutually corroborative, which validates the reliability of our experimental data. The power-law diffusive exponent $\alpha(\tau)$ is calculated as

$$\alpha(\tau) = \frac{d \ln \langle \Delta r^2(\tau) \rangle}{d \ln \tau} \tag{2}$$

The diffusive exponent $\alpha(\tau)$ indicates how fast these particles spread out over time, can change at short times due to local quirks. But at long times, $\alpha(\tau)$ settles into a constant value. In this work, we focus on the long-time limit of the diffusion exponent. such that the diffusive exponent $\alpha(\tau) = \alpha$ is constant. In Newtonian fluids, $\alpha = 1.0$, and probe particles follow Fickian diffusion. For non-Newtonian viscoelastic fluids, it is generally found that probe particles exhibit subdiffusive behavior such that $0 < \alpha < 1$. Mean-squared displacements (MSDs) of probe particles in condensates from h0 to h8 are shown in Fig. 2i.

Normally, particles spread out with a diffusive exponent $\alpha = 1$, called Fickian diffusion. At the beginning (h0), probe particles exhibit diffusive exponent $\alpha = 0.8054$, close to Fickian diffusion. By h8, $\alpha$ decreases significantly, approaching nearly 0, which suggests the formation of a network structure.

Moreover, we studied the linear viscoelastic moduli (i.e., Storage modulus G'($\omega$) and loss modulus G''($\omega$)) as a function of frequency $\omega$ in h0, h4 and h8 condensates via PTM. The storage modulus G'($\omega$) and loss modulus G''($\omega$) for condensates at hours h0, h4 and h8 are presented in Fig. 2j. At the initial time point (0 h), the loss modulus G'' exceeds the storage modulus G' across all measured frequencies, indicating that the condensates exhibit viscous behavior without the formation of an entangled network. As time progresses to h4, the condensates display characteristics of viscoelastic Maxwell fluids. Specifically, at low angular frequencies ($\omega$), G'' remains greater than G', reflecting a predominantly liquid-like response. However, at higher frequencies, G' surpasses G'', indicating a shift to elastic-dominated mechanical behavior. The transition between predominantly elastic and predominantly viscous behavior occurs at a critical angular frequency $\omega_c$, where G' equals G''. The Maxwell relaxation time $\tau_c$, can be determined by the critical frequency $\tau_c = 1/\omega_c$, which characterizes the condensate's response time to deformation. For deformations shorter than $\tau_c$, the condensates exhibit a predominantly elastic response, while for deformations longer than $\tau_c$, the response becomes primarily viscous, leading to droplet flow. At h8, the condensate shows a storage modulus G' greater than the loss modulus G'' across all frequencies, indicating a transition to elastic behavior. Overall, the storage modulus G' for all three time points remains below 10 Pa.

## Co-aggregation of enzymes and phase-forming scaffolds in aged condensates

We then investigated the structural changes occurring during liquid-to-solid phase transition of non-equilibrium condensates. At h0, both the enzymes and phase-forming scaffolds exhibited homogeneous distribution within the condensates, as evidenced by uniform fluorescent signals (Fig. 3a and b). However, over time, we observed fluorescent puncta indicating co-localization of the enzymes and phase-forming scaffolds, suggesting their co-aggregation (Fig. 3a and b). To further elucidate the composition of these aggregates, we performed a comparative extraction assay on the condensates (Fig. 3c). Particularly, the biomolecular condensates were separated by ultracentrifugation. The isolated biomolecular condensates were then sequentially treated with 1 M NaCl and 6 M guanidine hydrochloride. The solubilized components were resolved by SDS-PAGE analysis. We hypothesized that proteins would be released from the condensates upon dispersion in a high ionic strength buffer; conversely, if they form stable aggregates, extraction would only be achievable with 6 M guanidine hydrochloride. Indeed, most h0 condensates were readily solubilized in the high ionic strength buffer, releasing significant quantities of FIB1-GFP-RIAD (-0.94) and MenH-RIDD (-0.93) (Fig. 3d and e). Importantly, the fractions of extracted proteins using 1 M NaCl decreased over time. In contrast, the majority of FIB1-GFP-RIAD (-0.96) and MenH-RIDD (-0.94) within h8 condensates were only extractable by 6 M guanidine hydrochloride (Fig. 3d and e), indicating that both the phase-forming scaffolds and enzymes are incorporated into the protein aggregates. While prior studies have primarily focused on the structural alteration of phase-forming scaffolds, revealing the formation of fibers or aggregates due to condensate aging, our data underscore that client proteins, such as the enzymes, are also integrated into the protein aggregates, likely through interactions with the phase-forming scaffolds.

We subsequently examined the internal structure of the condensate using transmission electron microscopy (TEM). At h0, condensates appeared as a uniform electron-dense area (Fig. 3f, Supplementary Fig. 9 and Supplementary Fig. 10). Over time, we

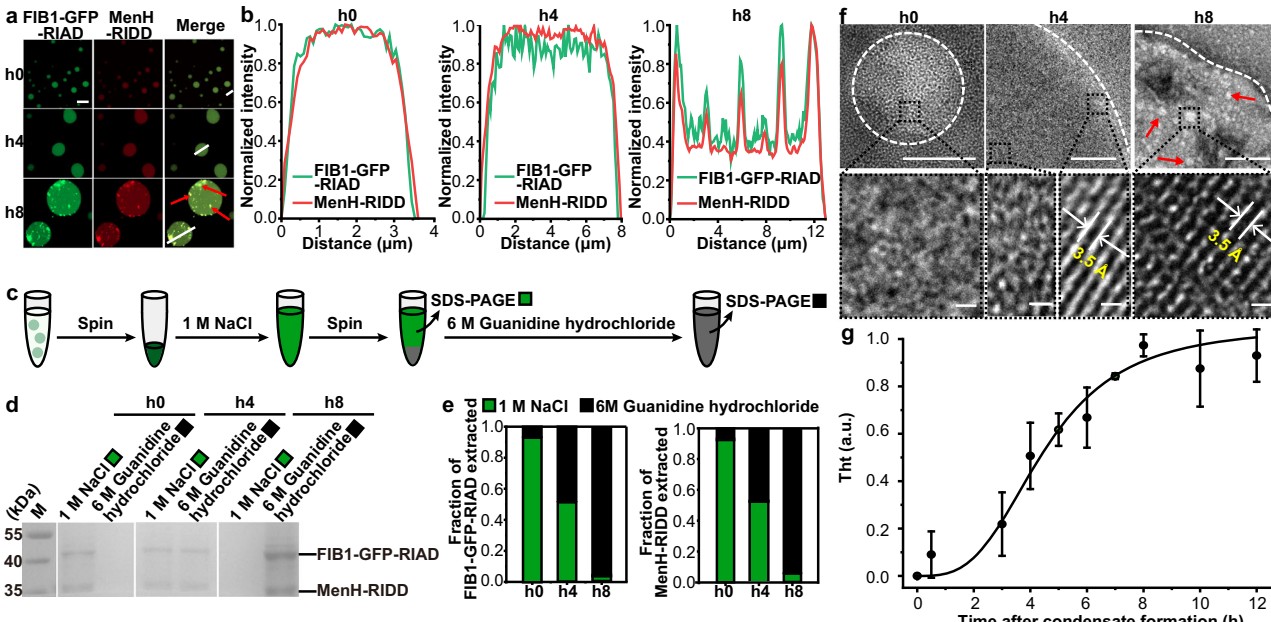

**Fig. 3 | Accumulation of protein aggregates of scaffold proteins and enzymes in aged condensates. a** Confocal images showing the onset and progression of protein aggregates in catalytic condensates. A 1:9 (v/v) ratio of Cy5-labeled to unlabeled MenH-RIDD was used to visualize the enzymes. The co-localization of enzyme and scaffold signals suggests co-aggregation of the proteins. Protein aggregates are indicated by red arrowheads. Scale bar, 10 μm. **b** Normalized fluorescent intensities along the white lines in **a** showing the co-localization of FIB1-GFP-RIAD and MenH fluorescent puncta. **c** Schematic of the comparative extraction assay. Proteins within the condensates were solubilized using a sequential treatment with different buffers (i.e., 1 M NaCl and 6 M guanidine hydrochloride), followed by SDS-PAGE analysis of the extracted components. SDS-PAGE analysis of proteins extracted from condensates using sequential treatments with varying buffers. **e** Quantification of the data in (**d**), illustrating the fractions of extracted proteins by different buffers over time. **f** TEM images of catalytic condensates at different stages. Particles with increased electron density are observed over time, indicating the formation of protein aggregates (top). Higher magnification images reveal an increase in crystalized areas exhibiting an approximately 3.5 Å lattice spacing characteristic of β-sheet structures (bottom). Protein aggregates are indicated by magenta arrowheads. The white dashed outline the edge of the condensates. Scale bars, 200 nm (top) and 0.5 nm (bottom). **g** Normalized Thioflavin T (ThT) fluorescence signal over time, demonstrating the accumulation of β-sheet structure within the condensates. The normalized fluorescence values were fitted to the Sigmoid function, with $R^2$ values > 0.99. Data are represented as mean ± SD from three independent experiments (*n* = 3). Source data are provided as a Source Data file.

observed particles with stronger electron density within the condensates, indicative of protein aggregate formation (Fig. 3f, Supplementary Fig. 9 and Supplementary Fig. 10). The discrepancy in the diameter of biomolecular condensates at h0 between TEM and confocal imaging is likely attributed to uranyl acetate staining during TEM preparation (Supplementary Fig. 11). Consistently, the PTM result also indicates the onset of subdiffusive behavior, as evidenced by a decrease in diffusive exponent α from 0.8054 to 0.0107 (Fig. 2i). This behavior is likely attributed to the formation of protein aggregates over time, which impede the thermal motion of probe particles. Higher-resolution TEM images revealed an increase in short-range ordered crystalline structures with a lattice spacing of approximately 3.5 Å (Fig. 3f), a characteristic periodicity of β-sheets[51]. By h4, although the interior remained disordered, crystallized areas began to develop at the edges of the condensates (Fig. 3f). The distinct difference between the interior and the surface of the condensates consistently corroborates recent reports that liquid-to-solid transitions originate at the surface of condensates[38,40]. Thiofavin T (ThT) staining further confirmed the emergence of these crystallized structures during condensate aging, evidenced by a gradual increase in fluorescence, which specifically binds to β-sheet structure (Fig. 3g). Previous studies have shown that the liquid-to-solid transition of condensates is closely related to the development of β-sheet structures[39,40,52]. Our data suggest that these short-range β-sheet structures may act as nucleation centers for the formation of protein aggregates, contributing to the solidification of the condensates. It is also noteworthy that recent simulations reveal that aging significantly impacts protein conformations within biomolecular condensates[53–55], such as disorder-to-order

structural transitions and the formation of kinked cross-beta sheets. Desolvation could promote a transition from a viscous to an elastic state by bringing sticky regions closer. In addition, our simulation results also showed that aging could affect the enzyme catalytic activity through aggregation (Supplementary Fig. 12 and Supplementary Data 1–6).

Interestingly, we observed comparable reaction rates for the free enzyme in solution across varying viscosities achieved through different sucrose concentrations (Supplementary Fig. 13), which mimic the more viscous environment inside condensate during aging[42]. This result suggests that restricted diffusivity alone is unlikely to influence the catalytic efficiency of the enzyme, at least within the range of viscosities we investigated. Together, these data demonstrate that condensate aging leads to the formation of protein aggregates containing both the phase-forming scaffolds and client enzymes, which in turn resulted in decreased enzyme activity, possibly due to reduced active site accessibility and altered conformations.

## Solidification of intracellular condensates impairs enzyme activity in living cells

We next investigated whether enzyme activity is compromised in living cells containing intracellular condensates. The ability of FIB1-GFP-RIAD to form condensates within the cytoplasmic environment was confirmed (Supplementary Fig. 14). Using mCherry as model client proteins, we demonstrated specific recruitment of cargo proteins via the RIAD-RIDD interaction (Supplementary Fig. 15). To monitor the enzyme reaction in living cells, we used a Renilla luciferase (Rluc)-mediated assay, which emits blue light (480 nm) during the oxidation

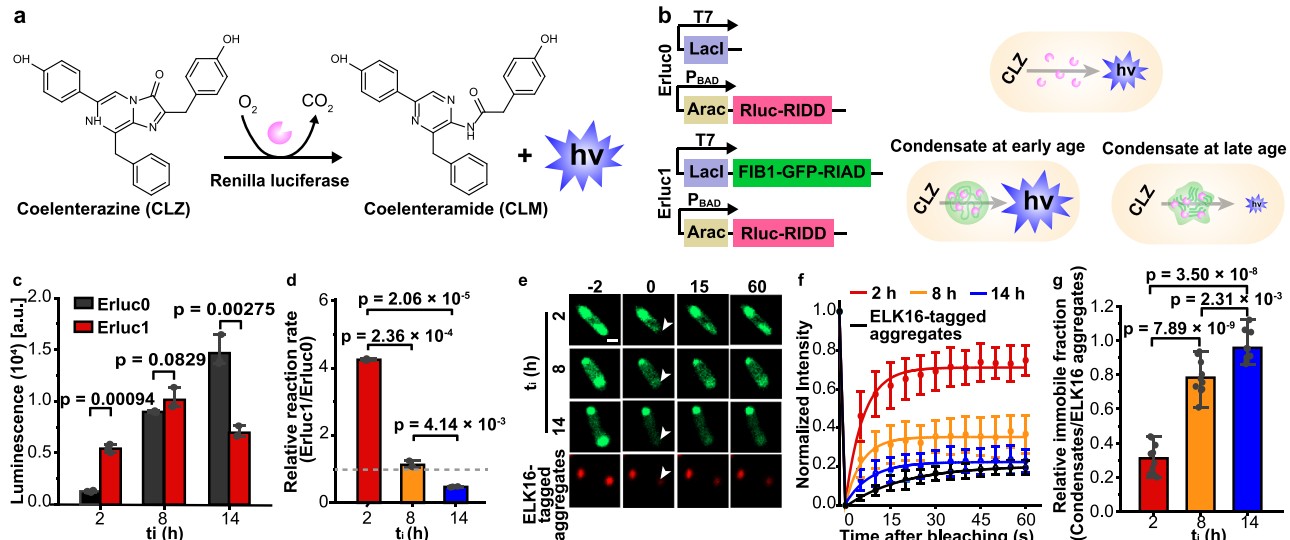

**Fig. 4 | Time-dependent enzyme activity in cells with intracellular condensates.** **a** Schematic representation of the Renilla luciferase (Rluc)-catalyzed reaction, which oxidizes coelenterzine (CLZ) to coelenteramide (CLM) with concomitant emission of blue light. **b** Genetic constructs used for the expression of RIDD-tagged enzyme (Rluc-RIDD) and the phase- forming scaffold (FIB1-GFP-RIAD), or the RIDD-tagged enzyme in its free form (left). The engineered strain of Erluc1, containing intracellular condensates, is expected to exhibit an enhanced reaction rate shortly after condensate induction but a diminished rate over time compared to Erluc0 (right). **c** Comparison of Rluc catalytic efficiency between Erluc0 and Erluc1 at various point $t_i$ following FIB1-GFP-RIAD induction. A.u. on the vertical axis refers to absorption units. **d** Comparison of relative reaction rates. The normalized reaction rates were calculated as the ratio of luminescence intensities between Erluc1 and

Erluc0. The gray dashed line serves as a visual reference indicating that a relative reaction rate of 1 suggests comparable reaction rates between Erluc1 and Erluc0. **e** In-cell FRAP analysis of the intracellular condensates showing a gradual decrease in fluorescent recovery over time. *E. coli* cells expressing mCherry-ELK16 serve as a control for a solid matter. **f** FRAP recovery curves of intracellular condensates at different time points. Scale bar, 1 μm. **g** Immobile fractions of intracellular condensates normalized to mCherry-ELK16 aggregates, plotted as a function of time, derived from **f**. For **c** and **d**, data are represented as mean ± SD from three independent experiments ($n = 3$). For **f** and **g**, data are represented as mean ± SD from ten condensates ($n = 10$) from three independent experiments. Source data are provided as a Source Data file.

of coelenterazine (CLZ) to coelenteramide (CLM) (Fig. 4a)[56]. We constructed two compatible plasmids: one encoding RIDD-tagged luciferase (Rluc-RIDD) under the arabinose promotor on a pBAD vector, and another containing the FIBI-GFP-RIAD under the T7 promotor on a pACYC-Duet1 vector, enabling orthogonal expression of the phase-forming scaffolds and enzymes (Fig. 4b). Co-transformation of these plasmids into *E. coli* generated the engineered strain Erluc1 (Fig. 4b), where the formation of intracellular condensates was confirmed via confocal microscopy (Supplementary Fig. 16). As a control, we constructed the strain Erluc0, which expresses only Rluc-RIDD (Fig. 4b). Luminescence intensity, indicative of catalytic efficiency, was monitored after substrate addition at defined times post-induction of FIB1-GFP-RIAD (denoted as $t_i$). At early stages after condensate formation ($t_i = 2$ h), Erluc1 displayed a 4.3-fold increase in luminescence compared to Erluc0 (Fig. 4c and d), suggesting enhanced catalytic efficiency in the presence of intracellular condensates. However, at $t_i = 8$ h, the luminescence difference between Erluc1 and Erluc0 diminished, reaching similar levels (Fig. 4c and d). Notably, at $t_i = 14$ h, Erluc1 exhibited a 2.1-fold decrease in luminescence relative to Erluc0 (Fig. 4c and d), indicating impaired enzyme activity in aged condensates. Western blot analysis confirmed comparable expression levels of Rluc-RIDD between Erluc1 and Erluc0 at all time points, ruling out differential enzyme expression as a contributing factor to the altered reaction rates (Supplementary Fig. 17).

To explore whether intracellular condensates solidified over time, we first performed in-cell FRAP experiments. ELK16-tagged mCherry-ELK16 served as a control for solid-like aggregates. ELK16, a self-assembling peptide, is known to drive insoluble aggregate formation when fused to target proteins[57]. Early-stage condensates ($t_i = 2$ h) exhibited a rapid fluorescence recovery within 20 s, indicating a high molecular mobility (Fig. 4e and f). In contrast, at $t_i = 14$ h, fluorescence recovery was markedly reduced, resembling the dynamics of mCherry-

ELK16 aggregates and suggesting a transition to a solid-like state (Fig. 4e, f). The relative immobile fraction of condensates normalized to mCherry-ELK16 aggregates increased from 0.31 to 0.96 over time (Fig. 4g), indicating that aged intracellular condensates exhibit a similar immobile fraction to that of mCherry-ELK16. These findings indicate that intracellular condensates are metastable and age into a dynamically arrested, solid-like state.

## Aged intracellular condensates accumulate protein aggregates and lose phase-separated barriers

To investigate the formation of protein aggregates within aged intracellular condensates, we examined *E. coli* strain Erluc1co-expressing Rluc-RIDD and FIB1-GFP-RIAD using correlative light electron microscopy (CLEM). Fluorescence imaging of FIB1-GFP-RIAD was achieved via an anti-GFP primary antibody coupled with a fluorescently labeled secondary antibody. Overlap of fluorescent signals with electron-dense regions confirmed that these regions correspond to synthetic biomolecular condensates (Fig. 5a and Supplementary Fig. 18). These findings strongly suggest that the electron-dense regions observed by EM represent synthetic intracellular condensates.

To achieve higher resolution and gain detailed insights into the internal architecture of the condensates, we employed electron tomography (ET). This approach improves upon the resolution of traditional two-dimensional (2D) EM by enabling three-dimensional reconstructions of complex cellular structures from tilt-series images. Consistent with our previous EM observations, tomography images of early-stage condensates ($t_i = 2$ h) revealed uniform electron-dense regions (Fig. 5b, Supplementary Fig. 18 and Supplementary Movie 1), confirming their identity as synthetic biomolecular condensates. At later time points ($t_i = 8$ h), however, protein aggregates emerged within the condensates (Fig. 5b, Supplementary Fig. 18 and Supplementary Movie 2). Tomographic images and their corresponding 3D

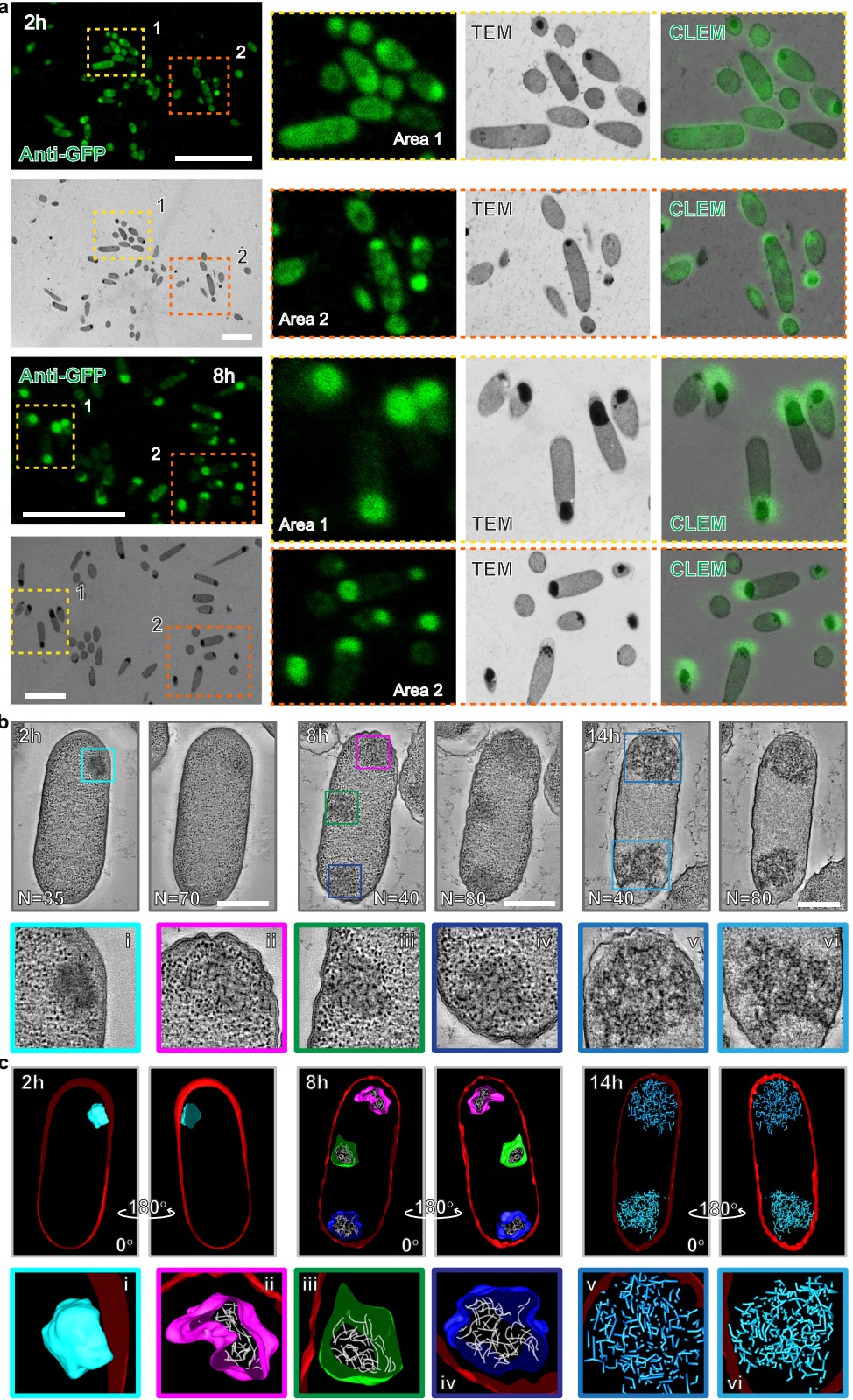

reconstructions revealed these aggregates as short, rod-like structures (Fig. 5b, c, Supplementary Fig. 18 and Supplementary Movie 2). Over time, these protein aggregates grew and interconnected, forming a network-like structure (Fig. 5b, c, Supplementary Fig. 18 and Supplementary Movie 3). These findings suggest that condensate solidification is driven by the formation of protein aggregate networks. Interestingly, the morphology of these aggregates differed from the dot-like aggregates typically observed in in vitro EM studies. This discrepancy likely arises from the distinct environmental conditions within living cells compared to those in vitro. Collectively, these results demonstrate that aged intracellular condensates, akin to their reconstituted in vitro counterparts, accumulate protein aggregates, resulting in condensate solidification and a concomitant decline in enzymatic activity.

**Fig. 5 | Electron microscopy images showing that aged intracellular condensates accumulate protein aggregates and lose phase-separated barrier.**
**a** Correlative light electron microscopy (CLEM) of *E. coli* strain expressing synthetic condensates. A representative confocal image overview is shown (upper left), with the correlated region further visualized by EM (bottom left). Regions highlighted in yellow and orange boxes are magnified in the right panels. Cells were fixed using high-pressure freezing and embedded in resins. Thin-sections (100 nm) were incubated with an anti-GFP primary antibody, followed by a fluorescently tagged secondary antibody. Confocal microscopy images were aligned with TEM images based on cell structures and distribution. Scale bars, 10 μm (confocal) and 4 μm (TEM). **b** Representative tomographic slice images overview showing the ultrastructure of *E. coli* cells containing intracellular condensates at different stages (top). High-magnification of electron tomographic images reveal detailed

condensate structures, with red arrows indicating protein aggregates (bottom). *E. coli* cells expressing intracellular condensates were fixed and examined using electron tomography. 250 nm thick TEM sections were imaged from multiple angles (± 60°) and aligned together using etomo software. Scale bars, 500 nm. **c** Three-dimensional (3D) models generated from tomographic reconstructions (top). Colored boxes indicate condensates at various stages and magnifications (bottom): (i), early-stage condensates ($t_i$ = 2 h); (ii–iv), middle-stage condensates ($t_i$ = 8 h) with protein aggregates; (v and vi), late-stage condensates ($t_i$ = 14 h) showing protein aggregates and cytosolic infiltration. Cell membranes are highlighted in red, protein aggregates in white (middle-stage condensates) and blue (late-stage condensates). The colored surfaces around the condensates represent pseudo-barriers between condensates and cytoplasm. All experiments have been repeated at least three times with consistent results.

Additionally, aged condensates exhibited a loss of their ability to function as isolated compartments distinct from the cytosolic environment, as evidenced by the infiltration of cytoplasm into the synthetic condensates (Fig. 5c, Supplementary Fig. 18 and Supplementary Movie 3). To further support this observation, we performed in vitro experiments using Cy5-labeled Dextron200, which is similar in size to ribosomes. In freshly formed condensates, Dextron200 did not exhibit significant enrichment within the condensate (Supplementary Fig. 19), indicating that the condensates initially retained their selective permeability. In contrast, aged condensates showed significant Dextron200 enrichment (Supplementary Fig. 19), indicating that their ability to exclude non-specific solutes was compromised, likely due to the altered chemical solvating properties of the condensates over time[58]. This loss of the pseudo-barrier underscores the failure of aged condensates to function as membraneless organelles, as they can no longer maintain a compartmentalized environment. Given that the internal environment of condensates can influence biomolecular activities[59], further investigation is necessary to explore how the altered permeability of condensates affects enzyme activities within these structures.

## Rescue of impaired enzyme activity through condensate aging inhibitors

To mitigate the impaired enzymatic activities observed in aged condensates, we explored the use of condensate aging inhibitors. Based on previous studies, we tested the biocompatible molecules ATP, glycine, and tryptophan for their ability to modulate condensate aging[60]. Remarkably, glycine preserved the spherical morphology of biomolecular condensates at h8, indicative of a liquid-like state. In contrast, tryptophan induced irregular, solid-like condensates as early as at h0, while ATP produced no discernible change in morphology compared to conditions without small molecules addition (Fig. 6a). Consistent with this, FRAP analysis revealed that ATP did not restore liquid-like properties within the condensates at h8 (Supplementary Fig. 20). However, glycine partially recovered the dynamic properties of the 8-hour-aged condensates (Fig. 6b). Furthermore, PTM analysis revealed a higher diffusive exponent α in the presence of glycine (0.3339) than in the absence (0.0107), indicating that glycine can inhibit condensate aging and thus lead to a more fluid-like property at h8 (Fig. 6c). Conversely, tryptophan impaired the dynamics of freshly formed h0 condensates, accelerating their solidification (Fig. 6d, e). These observations differ from earlier reports suggesting both ATP and tryptophan inhibit condensate aging[60]. We attribute this discrepancy to the differences in the phase-forming scaffolds used, which may respond uniquely to these molecules. Overall, these results demonstrate that small molecules can modulate the kinetics of condensate aging. Based on these findings, glycine and tryptophan were selected as inhibitors and promoters, respectively, for modulation of condensate aging in the following studies.

We next investigated whether condensate aging inhibitors could restore enzymatic activity in aged condensates at h8 (Fig. 6f). Two experimental groups were established: (1) h8$_{+Gly}$ (8-hour-aged catalytic condensates in the presence of glycine) and (2) h8$_{-Gly}$ (8-hour-aged catalytic condensates in the absence of glycine). The reaction rate in the h8$_{+Gly}$ group was 7.2 times higher than in h8$_{-Gly}$ (Fig. 6g). To confirm that this increase was due to inhibition of condensate aging rather than a direct effect of glycine on enzymatic activity, we compared the relative reaction rates normalized to those of a control solution containing equivalent enzyme and glycine levels but lacking condensate. Although both relative reaction rates were below one, the rate for h8$_{+Gly}$ was about 3.6-fold higher than that of h8$_{-Gly}$ (Fig. 6h). This indicates partial restoration of enzymatic activity in the h8$_{+Gly}$ system due to glycine. Similarly, we examined whether a condensate aging promoter could attenuate enzymatic activity in freshly formed condensates. Two experimental groups were established: (1) h0$_{+Trp}$ (0-hour-aged catalytic condensates in the presence of tryptophan) and (2) h0$_{-Trp}$ (0-hour-aged catalytic condensates in the absence of tryptophan). As expected, h0$_{+Trp}$ system exhibited significantly reduced reaction rates compared to h0$_{-Trp}$ (Fig. 6i and j). These results demonstrate that tryptophan accelerates condensate aging and inhibits enzymatic reactions within condensates. Collectively, these results establish that small molecules capable of modulating condensate aging can influence enzymatic reactions within non-equilibrium condensates in vitro.

We further explored the application of condensate aging modulators in living cells harboring intracellular condensates. Three experimental groups were established: (1) Erluc1$_{+Gly}$ (*E. coli* strain Erluc1 cultivated with glycine), (2) Erluc1$_{+Trp}$ (*E. coli* strain Erluc1 cultivated with tryptophan), and (3) Erluc1 (Erluc1 cultivated without supplementation) (Fig. 6k). Corresponding control groups cultivated under the same conditions but without condensates (Erluc0$_{+Gly}$, Erluc0$_{+Trp}$, and Erluc0) were included. Relative luminescence intensity among the experimental groups was compared by normalizing the luminescence intensity of each group to that of its respective control group. We found that the Erluc1$_{+Gly}$ exhibited significantly higher reaction rates compared to Erluc1, while the Erluc1$_{+Trp}$ displayed reduced relative reaction rates (Fig. 6l). These results align with in vivo FRAP analyses of catalytic condensates, where glycine inhibited aging and tryptophan promoted it (Fig. 6m). Further controls confirmed that cells with or without small molecules displayed similar expression levels of luciferase (Supplementary Fig. 21). Additionally, neither glycine nor tryptophan altered bacterial growth rates (Supplementary Fig. 22), indicating no cytotoxicity at the concentrations used. Together, these findings highlight the ability of glycine and tryptophan to modulate condensate aging and regulate enzymatic activity in intracellular condensates.

Additionally, we explored whether condensate aging could be modulated by other factors such as salt concentration, pH, and arginine. We found that elevated salt concentrations, reduced pH values,

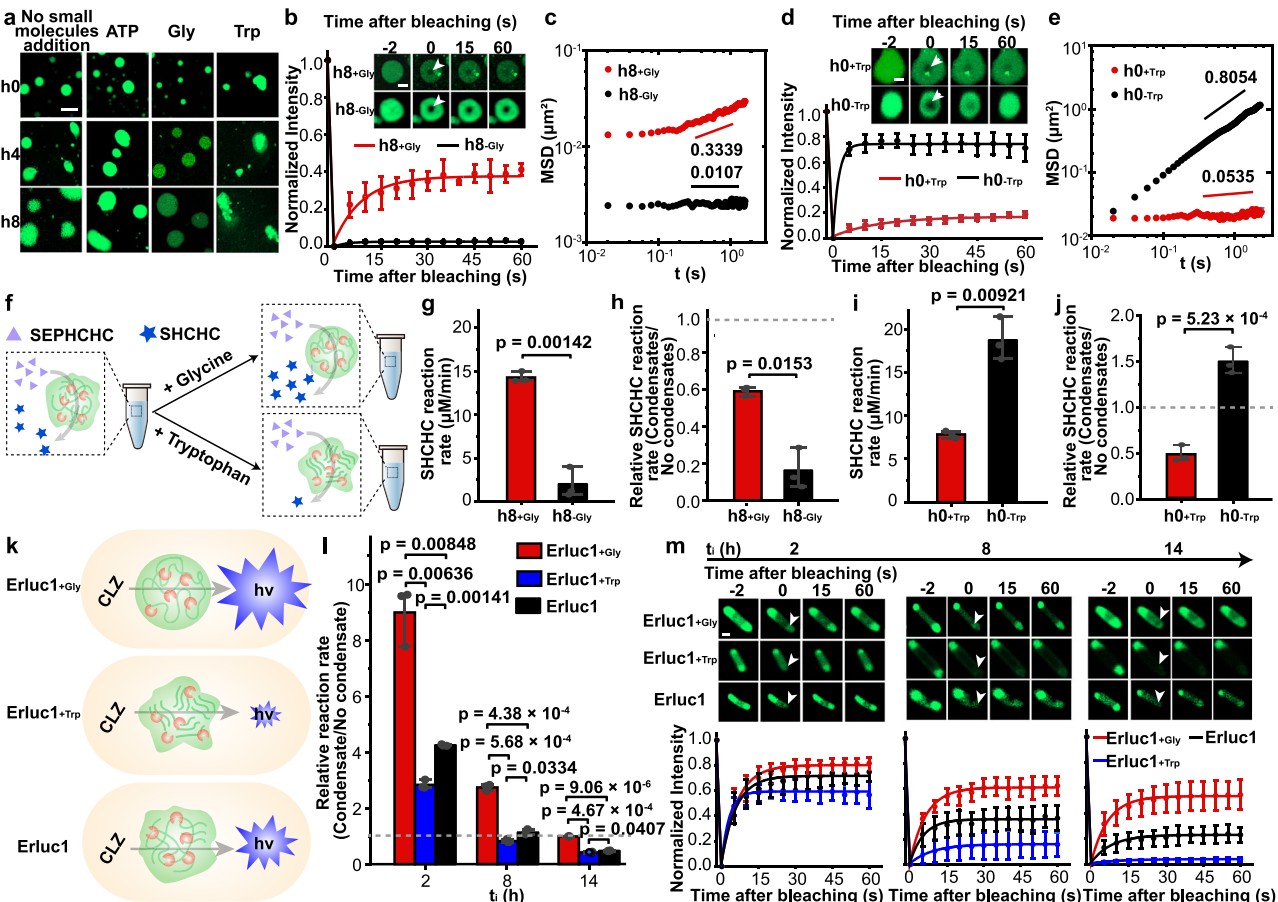

**Fig. 6 | Rescue of impaired enzymatic reactions in aged condensates via modulation of condensate aging. a** Confocal fluorescence image of catalytic condensate over time in the presence of 0.2 mM ATP, glycine, tryptophan, or in the absence of small molecules. Scale bar, 10 μm. **b** FRAP recovery curves for 8-hour-aged condensates with (h8$_{+Gly}$) or without glycine (h8$_{-Gly}$). The inset shows representative images of condensates before bleaching, at bleaching, and after bleaching. Scale bar, 2 μm. **c** Averaged MSDs $\langle \Delta r^2(\tau) \rangle$ versus lag time τ for probe particles in h8 condensates with or without glycine. **d** FRAP recovery curves of fresh (h0) catalytic condensates with (h0$_{+Trp}$) or without tryptophan (h0$_{-Trp}$). The inset shows representative images of the condensates before bleaching, at bleaching, and after bleaching. Scale bar, 2 μm. **e** MSDs versus lag time τ for probe particles in h0 condensates with or without tryptophan. **f** Schematic of the regulation of enzyme activity via modulation of the process of condensate aging. **g** Comparison of SHCHC reaction rates between h8$_{+Gly}$ and h8$_{-Gly}$. **h** Relative reaction rates of h8$_{+Gly}$

and h8$_{-Gly}$ normalized to a control system containing equivalent enzyme and glycine levels but lacking condensates. **i** Comparison of SHCHC reaction rates between h0$_{+Trp}$ and h0$_{-Trp}$. **j** Relative reaction rates of h0$_{+Trp}$ and h0$_{-Trp}$ normalized to a control system containing equivalent enzyme and tryptophan levels but lacking condensates. **k** Schematic of experimental design. **l** Relative reaction rates in *E. coli* with or without condensate aging modulators. **m**, In-cell FRAP analysis of intracellular condensates. Data are represented as mean ± SD from ten condensates ($n = 10$) from three independent experiments. Scale bar, 1 μm. For **b** and **d**, data are represented as mean ± SD from five independent experiments ($n = 5$). For **g**–**j**, and **l**, data are represented as mean ± SD from three independent experiments ($n = 3$). For **h**, **j**, and **l**, the gray dash line indicates a relative reaction rate of 1, signifying comparable rates between the systems containing condensate and free enzyme systems. Source data are provided as a Source Data file.

or supplementation with arginine delay condensate aging (Supplementary Figs. 23–27). Furthermore, we examined how salt concentration and pH influence reaction rates within condensates. Notably, we observed that impaired enzyme activities in aged condensates can be rescued by lower pH values and higher NaCl concentrations (Supplementary Fig. 28 and Supplementary Fig. 29). Together, these results indicate that multiple factors influence condensate aging, which can be leveraged to tune enzyme reaction rates within condensates. Further studies are required to investigate the generalizability of these factors in modulating the aging of other biomolecular condensates as different phase-forming scaffold interaction networks could yield distinct environmental sensitivities.

## Discussion

While the critical role of biophysical properties in biomolecular condensates for dictating the function of recruited proteins is increasingly well-recognized[29–34], our understanding of how these properties

evolve over time and influence enzymatic activity remains incomplete. Here, we develop a robust and modular synthetic condensate system, made from IDR, to elucidate the temporal effects of LLPS on enzymatic reactions. Despite the simplicity of this engineered system, the synthetic condensates recapitulate key features of naturally occurring biomolecular condensates. Their formation is driven by weak interactions among IDR (here, FIB1-GFP-RIAD), and enzymes are recruited and enriched through interaction with the scaffolds[6,61]. Additionally, the synthetic condensates exhibited restricted molecular diffusion, resistance to fusion, and loss of spherical morphology over time, akin to the aging or maturation of naturally occurring condensates[24,62,63].

By simultaneously tracking the biophysical evolution and enzymatic activities of catalytic condensates, we reveal that condensate aging adversely affects recruited enzymes, an aspect largely overlooked in prior studies. Although earlier work has highlighted mechanisms beyond mere increased local concentrations of enzymes that accelerate enzymatic reactions within condensates[12,16,18–21], we demonstrate that

such enhancements are transient. Shortly after formation, the synthetic condensates display liquid-like properties that facilitate elevated reaction rates (Figs. 1e–g, 4c and d). Over time, however, the condensates transition to a solid-like state, diminishing these enhancements and ultimately impairing enzymatic activity in late-stage condensates (Figs. 1e–g, 4c and d). Further, addition of small molecules that modulate condensate aging either rescued or inhibited enzymatic activity (Fig. 6g–j and l). Together, these findings suggest that enzymatic activities within non-equilibrium biomolecular condensates are highly time-dependent, with aging attenuating the favorable effects of LLPS. Given that biomolecular condensates by LLPS are vulnerable to undergo a further liquid-to-solid phase transition, our finds strongly support the importance of considering the effect of condensate aging on recruited enzymes when evaluating the functional consequences of naturally occurring condensates and designing synthetic biomolecular condensates for creating new materials and functions.

Our results suggest that the progressive decline in enzymatic activity within aged condensates stems from the formation of abnormal protein aggregates. While some studies have implicated stable aggregate formation in the solidification of condensates[4,24,26,35], others suggest a glass-like state characterized by minimal structural reorganization[34,42]. Using particle tracking microrheology (PTM), we observe a gradual transition from a viscous liquid to an elastic solid through a Maxwell fluid intermediate (Fig. 2h–j), indicating the presence of solid aggregates in aged condensates. Consistent with these observations, we find that the scaffold proteins and enzymes co-aggregate into insoluble aggregates in aged in vitro condensates (Fig. 3a, b, d and e). Importantly, electron tomography reveals similar protein aggregates accumulating in aged intracellular condensates (Fig. 5b and c), corroborating our in vitro findings (Fig. 3f). These data suggest that co-aggregation of enzymes with scaffold proteins restricts access to enzymatic active sites, thereby impairing activity. Solidification of biomolecular condensates has been previously linked to disrupted biological processes through sequestration of key components[64,65]. For example, solid-like condensates of TDP43-SEC16A at endoplasmic reticulum exit sites (ERES) trap COPII complexes, impairing ER-to-Golgi transport[65]. However, whether enzymes are merely trapped in IDR-driven protein aggregates or themselves undergo structural alterations due to condensate aging remains unclear. Consistent with the latter hypothesis, prolonged LLPS has been shown to induce changes in protein conformation[66]. Future structural studies focusing on isolating and characterizing enzymes from condensates will provide atomic-level insights into how protein aggregation impacts enzymatic activity. Additionally, while small-molecule reactions appear unaffected by restricted molecular diffusion under our experimental conditions, further investigation is required to assess the mobility of biomacromolecules (e.g., RNA) as substrates, where diffusion-limited effects may be more pronounced.

Another key observation from our study is the loss of the phase-separated barrier in late-stage condensates, as evidenced by ribosome infiltration (Fig. 5c). A hallmark feature of biomolecular condensates functioning as organelle analogs is the presence of a pseudo-phase-separated barrier that excludes cytosolic components, thereby creating an optimal environment for enzymatic catalysis[58]. Previous studies have shown that condensates can exhibit distinct pH levels and polarity, facilitating their biochemical roles[67,68]. The loss of this barrier in aged condensates suggests functional deterioration, as they can no longer sustain the independent and specialized reaction conditions required for optimal activity. A systematic investigation of how the chemical environment within condensates evolves over time will provide a more comprehensive understanding of aging-associated dysfunctions.

In summary, we have identified condensate aging as a critical factor influencing enzymatic activity in non-equilibrium biomolecular condensates. Beyond the established mechanisms governing enzyme

activity in phase-separated systems, we show that aging further modulates enzymatic performance over time. These findings enhance our understanding of the functional consequences of LLPS and provide a foundation for designing synthetic condensates optimized for sustained enzyme functionality. As metastability is an inherent property of most LLPS-derived condensates, future studies should explore the cellular mechanisms that maintain condensates in non-equilibrium states with favorable biophysical properties. Such research will not only advance our understanding of natural condensate regulation but also inspire the development of synthetic condensates engineered to resist aging and maintain functionality over extended timescales.

## Methods
### Cloning
*E. coli* DH5α was employed for cloning. Antibiotics were used at the following concentrations for selection: 50 µg/mL carbenicillin, 50 µg/mL kanamycin, 50 µg/mL spectinomycin, and 35 µg/mL chloramphenicol. The genes encoding mCherry, RIDD, RIAD, and MenH are maintained as stocks in our laboratory. Codon-optimized DNA encoding FIB1-GFP-RIAD was synthesized and cloned into pET28a by Genewiz. Additionally, a sequence containing Rluc-RIDD was also synthesized and subsequently cloned into pBAD by Genewiz. Plasmids were constructed using Gibson assembly. Verification of the sequences was performed via Sanger sequencing by Genewiz. For detailed information regarding the plasmids, strains, primers, and amino acid sequences, please refer to Supplementary Tables 1–5.

### Cargo protein recruitment assays
To investigate cargo protein recruitment, a mixture of 1 µM cargo proteins (mCherry, FIB1$_{0.5}$-mCherry, FIB1-mCherry, RIDD-mCherry, or MenH-RIDD) and 6 µM FIB1-GFP-RIAD was prepared and incubated in a condensate formation buffer (20 mM Tris, 150 mM NaCl, 0.5 mM AEBSF, 0.1 mM DTT, 0.1% Triton X-100, 10% glycerol, pH 7.4). For the recruitment experiments of MenH-RIDD, a 1:9 (v/v) ratio of cyanine 5 N-hydroxysuccinimide (Cy5-NHS, Macklin, Cat. No.: C763657) labeled to unlabeled MenH-RIDD was employed to minimize the adverse effect of protein labeling on proteins. Images were collected following an incubation at room temperature for 30 min to facilitate encapsulation of cargo proteins. For fluorescence imaging by the LSM980 microscope, the mCherry and GFP signals were excited using a 594 nm laser and a 488 nm laser, respectively, while the Cy5 signal was excited with a 639 nm laser. Emission signals were recorded for mCherry (600–670 nm), GFP (510–560 nm) and Cy5 (660–760 nm). The partitioning of cargo proteins into the condensate was quantified by calculating the ratio of mCherry fluorescence intensity (or Cy5 for MenH-RIDD) within the condensate to that outside the biomolecular condensate.

### MenH catalyzed reaction in vitro
All enzymatic assays were performed in triplicate. Reaction conditions for the MenH-catalyzed reaction were based on our previous report[69,70]. In brief, the MenH substrate, SEPHCHC, was synthesized enzymatically from chorismate using the isochorismate synthase MenF and SEPHCHC synthase MenD, as previously described[70]. In brief, a reaction mixture containing 500 µM chorismate, 2 µM MenF, 1 µM MenD, 1.5 mM Mg$^{2+}$, was prepared in the condensate formation buffer, supplemented with 1 mM 2-ketoglutarate and 10 µM ThDP. This mixture was incubated at room temperature for 20 min. For reactions involving biomolecular condensates, 6 µM FIB1-GFP-RIAD and 1 µM MenH-RIDD were incubated at room temperature in condensate formation buffer for varying durations to generate catalytic condensates at different stages (h0, h4, and h8). For control reactions devoid of biomolecular condensates, equivalent amounts of enzyme were incubated under the same conditions but without the addition of FIB1-GFP-RIAD. Enzymatic reactions were initiated by the addition of 50 µM

SEPHCHC at room temperature. The formation of MenH's product, SHCHC, was monitored via its specific absorption at 293 nm using a Synergy HTX multimode plate reader (Agilent). The quantity of SHCHC was analyzed using a standard curve (Supplementary Fig. 30). For enzymatic reactions involved tryptophan, the absorbance values were corrected by subtracting the corresponding values obtained from a blank reaction buffer containing equivalent tryptophan but without enzyme.

For enzyme kinetic experiments, initial reaction rates of MenH at varying substrate concentrations were determined. The initial rates of MenH were plotted as a function of the corresponding SEPHCHC concentration. Apparent kinetic constants $k_{cat}$ and $K_M$ were determined by fitting the kinetic profiles to the Michaelis-Menten equation using OriginPro 2021 (v9.85.204).

## Fluorescence recovery after photobleaching (FRAP)

To investigate the dynamics of catalytic condensates, 6 μM FIB1-GFP-RIAD was mixed with 1 μM MenH-RIDD (10% Cy5-labled MenH-RIDD) and incubated for varying durations to produce condensates with different extents of aging. For experiments involving small-molecule modulators of condensate aging, catalytic condensates were prepared in the presence of 0.2 mM glycine or tryptophan. All FRAP experiments were performed at room temperature using a 63×/1.40 oil immersion objective on an LSM980 confocal microscope. A volume of 5 μL of in vitro-prepared catalytic condensates was applied to glass slides and covered with a 20×20 mm coverslip treated with 10% Pluronic F-127. The formation of condensates was confirmed by a 0.2% 488 nm laser. Regions of interest (ROIs) with diameters of approximately 2 μm (for in vitro experiments) were selected at the center of the condensates. The ROI was then photobleached using a 100% 488 nm laser for 2 s over 10 iterations to investigate the dynamics of phase-forming scaffold. Recovery following photobleaching was monitored for 60 s at 5-second intervals. Fluorescence intensities during FRAP were obtained by ZenPro software (Zeiss). To compensate for the bleaching effects during time-lapse imaging, the fluorescent intensities of ROI at time $t$ ($I_{(t)}$) was corrected against the fluorescent intensity of a reference non-bleached condensate (or an unbleached area within the same condensates but distant away) using following equations[71,72]:

$$C_f = \frac{R_i}{R_{(t)}} \tag{3}$$

$$I_{cor(t)} = C_f \times I_{(t)} \tag{4}$$

where $C_f$ is the correction factor, $R_i$ represents the initial fluorescent intensity of the reference non-bleached area, and $R_{(t)}$ is the fluorescent intensity of the reference non-bleached area at time $t$, the term $I_{cor(t)}$ denotes the corrected fluorescent intensity (corrected for unintentional bleaching) of the ROI at time $t$. $I_{cor(t)}$ was scaled to 0–1 range according to the following equation[71,72]:

$$I_{n(t)} = \frac{I_{cor(t)} - I_{min}}{I_i - I_{min}} \tag{5}$$

where $I_{n(t)}$ indicates the relative fluorescent intensity of ROI at time $t$, $I_{min}$ signifies the minimum fluorescence intensity of the ROI (recorded at 0 s after bleaching), and $I_i$ denotes the initial fluorescent intensity of ROI before bleaching.

The immobile fraction ($F_{im}$) of the condensates was determined using the following equation[73]:

$$F_{im} = \frac{I_i - I_\infty}{I_i - I_{min}} \tag{6}$$

where $I_\infty$ represents the final fluorescence intensity value after recovery. Based on this, mobile fraction ($F_m$) of catalytic condensates can be calculated by:

$$F_m = 1 - F_{im} \tag{7}$$

To calculate the halftime of recovery ($\tau_{1/2}$), the kinetic profiles were then fitted to the signal exponential model in OriginPro 2021 (v9.85.204)[74]:

$$I_{n(t)} = F_m \left( 1 - e^{-\frac{(\ln 2)t}{\tau_{1/2}}} \right) \tag{8}$$

Apparent diffusion coefficients ($D_{app}$) of catalytic condensates can be calculated with[71,75]:

$$D_{app} = \frac{0.22 r^2}{\tau_{1/2}} \tag{9}$$

where r refers to the bleaching radius of the ROI obtained by ZenPro software.

For FRAP analysis of the enzymes in the condensates, samples were excited with a 30% 639 nm laser. ROIs were chosen at the center of the condensates and photobleached using a 100% 639 nm laser for 2 s.

## Resistance of condensates to 1,6-hexanediol (1,6-HD) and dilution

1,6-HD experiments were conducted using solutions containing 6 μM FIB1-GFP-RIAD and 1 μM MenH-RIDD (10% Cy5-labeled MenH-RIDD) in condensate formation buffer. These solutions were incubated for varying durations (0, 4, and 8 h) before the addition of 10% (w/v) 1,6-HD (Sangon, Cat. No.: A601513-0100). Following the 1,6-HD treatment, the samples were allowed to equilibrate for 10 min prior to imaging with a Zeiss LSM980 microscope. The numbers of condensates before and after 1,6-HD treatment were determined using the Particle Analysis plugin in FIJI (v1.53f51). For dilution experiments, the same solutions were imaged before and after being diluted 20-fold with condensate formation buffer.

## Particle tracking microrheology (PTM)

Amino-modified fluorescence microspheres (Zhongkeleiming, Cat. No.: PSRF00200), with a radius of 100 nm, were diluted 500-fold. To this dilution, 6 μM FIB1-GFP-RIAD and 1 μM MenH-RIDD were added, and the mixture was incubated at room temperature for a specified duration of 0 h, 4 h, or 8 h. A 10 μL aliquot of the mixture was placed onto a 35 mm confocal dish with a 15 mm circular glass cover at the bottom (Beyotime) and sealed with 200 μL dimethylsiloxane fluid with viscosity of 1 Pa·s. PTM experiments were performed at room temperature using a 40×/0.8 water immersion objective. Imaging was performed on a Nikon SD-SORA confocal microscope equipped with a CSU-X1 spinning disk, utilizing a 561 nm laser for microspheres detection and a 488 nm laser for visualizing condensates. Images were taken by sCMOS camera (Nikon) at a rate of 50 frames per second for a total of 200 frames, generating video data at a resolution of 1024 × 1024 pixels. The center-of-mass positions of tracer particles are determined using a custom MATLAB R2024a (v24.1), and the particle positions across consecutive video frames are linked to generate trajectories[76–78]. Subsequently, the mean square displacements (MSDs) of the microspheres were calculated based on the recorded trajectories.

The viscoelastic moduli of a sample can be determined from the measurement of using the generalized Stokes−Einstein relation (GSER) under the assumption that probe particles are in a continuum and particle inertia is negligible[79,80]. The complex modulus $G^*(\omega)$ is given

by:

$$G^*(\omega) = G'(\omega) + iG''(\omega) \qquad (10)$$

For discrete MSD data acquired from the experiment, can be approximated as:

$$G^*(\omega = 1/\tau) = \frac{2k_B T \exp\left[\frac{i\pi\,\alpha(\omega)}{2}\right]}{a\left\langle \Delta r^2\left(\frac{1}{\omega}\right)\right\rangle \Gamma[1+\alpha(\omega)]} \qquad (11)$$

Where $k_B$ is the Boltzmann constant, a is probe (bead) radius, $\Gamma$ is the $\Gamma$-function and the angular frequency $\omega = 1/\tau$.

## Comparative extraction assay

Mixtures of 6 µM FIB1-GFP-RIAD and 1 µM MenH-RIDD were incubated in condensate formation buffer for varying durations (0, 4, and 8 h). Catalytic condensates at different time scales were collected by centrifugation at 13,000 × g for 3 min. The collected condensates were solubilized by 1 M NaCl for 30 min. After centrifugation at 13,000 × g for 3 min, the supernatants were then subjected to sodium dodecyl sulfate-polyacrylamide gel electrophoresis (SDS-PAGE). The remaining pellets were then solubilized by 6 M guanidine hydrochloride and then analyzed by SDS-PAGE. Gels were imaged by a ChemiDoc XRS+ System (Bio-Rad) in Coomassie Blue mode. Gel images were analyzed by FIJI.

## Transmission electron microscopy (TEM)

TEM was conducted on a JEM-F200 electron microscope (JEOL). Sample preparation followed an established protocol[69]. In brief, copper grids (Twinsolution) coated with formvar carbon films were subjected to glow discharge to increase their hydrophilicity. A 10 µL aliquot of catalytic condensates (6 µM FIB1-GFP-RIAD and 1 µM MenH-RIDD in condensate formation buffer), incubated for varying durations, was placed onto the treated copper grid and allowed to sit for approximately 10 min at room temperature. Excess sample was blotted away using filter paper. The grid was then immersed in 10 µL of 2% uranyl acetate staining solution for about 5 min, after which excess staining solution was removed with filter paper. The air-dried samples were observed on a TEM at a working voltage of 200 kV. Lattice spacing within catalytic condensates were measured using Gatan DigitalMicrograph (v3.6.1).

## Thioflavine T (ThT) assay

ThT fluorescence assay was conducted at room temperature. Samples of 10 µM ThT (Aladdin, Cat. No.: T168914), 6 µM FIB1-GFP-RIAD and 1 µM MenH-RIDD in the condensate formation buffer were added into in a 96-well plate (Corning). ThT fluorescence intensity was measured over time using a Synergy HTX multimode reader (Agilent), with excitation at 405 nm and emission at 480 nm. Prior to each measurement, the plate was shaken to ensure complete mixing. An identical solution without 6 µM FIB1-GFP-RIAD was employed as a control. The ThT signal for samples was corrected by subtracting the corresponding values obtained from a condensate formation buffer containing equivalent catalytic condensates but without ThT. The normalized ThT fluorescence intensity was fitted to a sigmoid function using OriginPro 2021 (v9.85.204)[26]:

$$I_{(t)} = I_0 + \frac{I_{max} - I_0}{1 + e^{-k(t - t_{1/2})}} \qquad (12)$$

where t is the time after condensates formation, $I_{(t)}$ is the normalized ThT fluorescence intensity at a particular time point $t$, k is the growth rate of β-sheet content, and $t_{1/2}$ is the halftime of the ThT fluorescence intensity that has reached its maximum value. Three technical replicates were performed in each assay.

## All-atom molecular dynamics (MD) simulations

FIB1-GFP-RIAD and MenH-RIDD dimer structures were modeled using Chai-1, with the best pLDDT-scoring structure refined by a 100 ns MD simulation. Molecular docking of SEPHCHC to MenH was performed using Autodock Vina. Three systems were prepared: FIB1-GFP-RIAD/MenH-RIDDs/SEPHCHC complex, MenH-SEPHCHC complex (using docked SEPHCHC and MenH crystal structure 4MYD), and apo-MenH. Three parallel 500 ns MD simulations were conducted for each system using AmberTools (v23) with the Amber14SB force field on a Linux HPC cluster. Systems were solvated in TIP3P water and neutralized with 0.15 M NaCl. Simulations involved energy minimization, heating (100 ps), NVT equilibration (10 ns), NPT equilibration (10 ns), and production MD (500 ns at 300 K and 1 atm) using a 2 fs timestep with hydrogen bond constraints, Langevin thermostat, and isotropic pressure scaling. Energies and coordinates were saved every 10 ps; trajectories were analyzed with cpptraj (AmberTools v23) and PyMOL (v2.6). Full methods details can be found in the Supplementary Information.

## Rluc-catalyzed reaction in living cells

*E. coli* cells expressing Rluc-RIDD under the control of arabinose and FIB1-GFP-RIAD under the T7 promoter were constructed. The cells were initially cultured overnight in 6 mL of LB medium containing appropriate antibiotics at 37 °C. The overnight culture was then inoculated 1:100 in 50 mL LB media and grown at 37°C until reaching an $OD_{600}$ of approximately 0.4. Induction of enzyme was initiated by the addition of 0.3% (w/v) arabinose. When the $OD_{600}$ reached around 0.6, 1 mM isopropyl β-D-1-thiogalactopyranoside (IPTG, Aladdin, Cat. No.: I274316) was added to induce the expression of FIB1-GFP-RIAD. At specified time points post-IPTG addition (2, 8, and 14 h), 1 mL of cell culture was harvested by centrifugation. The resulting cell pellets were washed three times with 1 mL PBS and resuspended in 500 µL PBS. 196 µL of the resuspended sample was transferred to a 96-well plate and allowed to settle for 5 min at room temperature. The reaction was initiated by adding 4 µL of a 3 mM substrate coelenterazine solution. Luminescence was recorded immediately after substrate addition using a Synergy HTX multimode reader (Agilent) at room temperature.

## Correlative light and electron microscopy (CLEM)

*E. coli* samples were fixed with high pressure freezing and embedded in HM20 resin. Thin sections (90 to 150 nm) were produced with a UC7 ultramicrotome (Leica Microsystems) and then collected on nickel slot grids (Electron Microscopy Sciences, Cat. No.: EMS2010-Ni) coated with 0.75% formvar solution. Immunofluorescence labeling and CLEM imaging were carried out according to the previous report[81]. The dilution of primary GFP antibody (Rockland, Cat. No.: 600-401-215, Lot. No.: 35649) is 1:20 and Alexa Fluor™ 488 conjugated goat anti-rabbit secondary antibody (Invitrogen, Cat. No.: A-11008, Lot. No.: 2622404) is 1:200. The sections were stained with uranyl acetate (UA) and lead citrate (LC) to enhance the contrast before TEM imaging. Confocal and TEM images were correlated and aligned based on the section location on the EM grid and cell outlines inside the section.

## Thin-section TEM analysis

For TEM sample preparation, high-pressure freezing, freeze substitution, resin embedding, and ultramicrotomy were performed as described previously[82,83]. In brief, enzyme expression was induced by 0.3% arabinose when the *E. coli* cultures grew to an $OD_{600}$ of 0.4 in LB. FIB1-GFP-RIAD expression was induced when $OD_{600}$ reached -0.6 by adding 1 mM IPTG. Cells were incubated at 37 °C with shaking at approximately 200 rpm for specific durations. Cultures were harvested by centrifugation at 6000 × g for 10 min and then rapidly frozen with an EM ICE high-pressure freezer (Leica). For freeze substitution, the samples were substituted with 2% osmium tetroxide in anhydrous acetone and maintained at −80 °C for 24 h using an AFS2 temperature-controlling system (Leica). Subsequently, the samples were subjected

to three washes with precooled acetone and gradually warmed to room temperature over a period of 60 h. Infiltration with increasing concentrations of EPON resin mix (50% Epon resin monomer, 15% dodecenyl succinic anhydride, and 35% nadic methyl anhydride) was carried out at room temperature. The samples were then transferred into tin foil molds and polymerized by curing at 60 °C for 2 days. The embedded samples were sectioned into 90 nm-thick slices using an ultramicrotome (Leica, Germany). Micrographs were acquired using a Hitachi H-7650 TEM operating at 80 kV, coupled with a charge-coupled device (CCD) camera.

### 3D electron tomography

Electron tomography was conducted using a 200 kV Tecnai F20 electron microscope (FEI) following previously established procedures[84,85]. In brief, the tilt images were obtained from 250-nm-thick sections across a range of −60° to 60°, with 1.5° increments, while the grid was rotated by 90° for the collection of the other axis of the tilt image stack. Dual-axis tomograms were generated by utilizing pairs of image stacks with the *etomo* program of the IMOD software (v.4.11.25). The contours of cell structures were manually delineated and subsequently meshed using the *3dmod* program within the IMOD software suite.

### Statistics and reproducibility

All experiments including blots and microscopy images reported in this study have been performed with biological replicates at least three times, yielding consistent results. Data are presented as means ± SD (standard deviation) of the mean with the number of biological replicates (n) derived from independent experiments provided in the figure legends. All significance was calculated by one-sided t-test and the exact *p* value was represented in the figure.

### Reporting summary

Further information on research design is available in the Nature Portfolio Reporting Summary linked to this article.

## Data availability

Unless otherwise stated, all data supporting the results of this study can be found in the article, supplementary, and source data files. Source data are provided with this paper.

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

## Acknowledgements

This work was financially supported by grants from the National Natural Science Foundation (no. 22178046), the Fundamental Research Funds for the Central Universities (no. DUT23YG110), and the Fundamental Research Funds for the Central Universities (no. DUT25YG104) to W.K., the National Science Foundation (no. U22A20424), Dalian Municipal Program for Outstanding Young Scientific and Technological Talent (no. 2021RJ03), the Ningbo Natural Science Foundation (no. 2022J013) and the Ningbo Municipal Public Welfare Science and Technology Foundation (no. 2024S004) to C.X., the Southern Polytechnic College of Engineering and Engineering Technology (SPCEET) faculty scholarship and Kennesaw State University (KSU) Interdisciplinary Seed Grants to B.L., and the Research Grant Council of Hong Kong (no. GRF14113921, no. GRF14109222, no. GRF14110823, no. GRF14113424, N_CUHK462/22, and C4014-23GF) to B.-H.K. We thank Dr. Hongbo Xie from DUT Instrumental Analysis Center for the analysis of TEM data. We acknowledge the assistance of DUT Core Facilities of the School of Bioengineering.

## Author contributions

C.X., W.K., and B.L. conceived the project. W.K. designed the experiments of the study. Z.W. performed the experiments. B.L. and X.H. performed PTM. J.M., H.Q., and J.L. carried out CLEM, ET, and thin-section TEM. J. Wu and S.W. performed simulations. J. Wang helped perform the experiments. C.X., B.L., and J.M. helped analyze the data. W.K. wrote the manuscript, which was edited and approved by all authors. B.L., J.M., B.K,. and C.X. helped summarize and give suggestions for the manuscript.

## Competing interests

The authors declare no competing interests.
