## [Transparent Peer Review file · Nature Communications]

Time-dependent catalytic activity in aging condensates

Corresponding Author: Professor Bo Li

Version 0:

Reviewer comments:

Reviewer #1

(Remarks to the Author)

The study by Kang et al. explores the impact of condensate aging on enzymatic activity using the engineered condensates from the intrinsically disordered region of fibrillarin paired with complementary interaction domains (RIAD and RIDD). Initially, these condensates enhance enzyme performance by concentrating both the enzyme and its substrate, which accelerates reaction rates. However, as the condensates age, they transition from a liquid-like to a solid-like state, resulting in a decline in enzyme activity. The authors support these findings with a range of techniques, including FRAP, particle tracking microrheology, TEM, electron tomography, and in vivo assays in *E. coli*.

This study is an important piece of information for understanding the connection between catalytic activity within condensates and their rheology, especially the aging process in vivo. I appreciate that the authors first conducted a significant study on in vitro condensates to understand the rheological transition during the aging process and then proceeded to a living *E. coli* system. However, although the material properties of condensates in *E. coli* show consistent behavior with the in vitro study, the actual regulatory mechanism of rheology and its consequences in the *E. coli* system remain elusive. My comments are listed below.

First, the authors claim that the condensate ages through an intermediate Maxwell fluid phase: before this regime, the condensate is liquid-like, and after it, the condensate is solid-like. However, this claim is not entirely accurate because the frequency range probed by the particle tracking experiment is inherently limited by the experimental time scale and resolution. It is important to note that the reported material properties apply only to the specific experimental time scale or frequency range used. For example, there is no guarantee that an h0 condensate would not exhibit Maxwell fluid behavior at higher frequencies (i.e., on smaller time scales). I suggest probing a wider range of frequencies or, at the very least, explicitly acknowledging this limitation.

Second, the authors claim that restricted diffusivity alone is unlikely to affect the enzyme's catalytic efficiency. However, the viscosity change shown in Figure S9 is quite minor. Studies such as Jawerth et al. (2020, *Science*) suggest that viscosity changes during aging can span several orders of magnitude. Therefore, I recommend investigating a wider range of viscosity changes and comparing them to the reaction rate.

Third, since various mechanisms could be speculated regarding how *E. coli* might modulate condensate rheology, I recommend comparing the results to different types of modulation, such as altering salt concentration and pH, at least in vitro. It would be interesting to see if these environmental changes lead to similar effects on aging and catalytic activity.

Finally, it would be valuable to extend the experiments and discussion to explore how aging affects the conformations of both the scaffolds and the client enzyme, thereby impairing catalytic activity. Consulting with computational studies might help to assess the possible structural modification during the aging process: Garaizar et al. (*PNAS*, 2022), Takaki and Thirumalai (*PNAS*, 2024), Biswas and Potoyan (*PTXLife*, 2024), etc.

Below are minor comments.

One cannot determine a "solid-like network" structure solely from rheology experiments (e.g., as mentioned in lines 312 and 320). I recommend refraining from structural "network" references unless there is definitive evidence from the rheology data that supports such a claim. The same problem applies to the claim "weak gels" in line 332.

Another minor comment is that it would be valuable to include a discussion about the role of surface tension and how it relates to the ability of condensates to maintain isolated compartments.

Although I am not an expert in biochemistry and cannot fully evaluate the biochemical details of the experiments, I found the study to be well executed and a valuable contribution to the literature on condensate aging. I believe that addressing the suggested comments will further enhance the accuracy and significance of the presented work.

Reviewer #2

(Remarks to the Author)

The manuscript by Kang et al. presents several results obtained through the application of TEM techniques, including negative staining, correlative light and electron microscopy (CLEM), and TEM tomography on sections of resin-embedded bacteria. The presented results contain several issues that need to be addressed before the manuscript can be considered for publication.

Negative Staining

Negative staining was used to image in vitro assembled condensates, as illustrated in Figure 3f. This image appears intended to provide higher-resolution views of the protein aggregates shown in Figure 3a. However, I find it difficult to correlate the sizes of condensates between Figure 3a and Figure 3f. The scale bar in Figure 3f (0h) corresponds to 200 nm, and the condensate shown is approximately 300 nm in diameter. In contrast, judging from the scale bar in Figure 3a (10 microns) and the data in Figure S14, where the average condensate diameter is reported as 4 microns. To clarify this discrepancy, I suggest including a series of low-magnification EM images displaying multiple condensates for each condition (0h, 4h, 8h) in the Supplementary Materials.

Additionally, the field of view of the magnified insets in Figure 3f is too small to reliably correlate the indicated features with those in the upper-row images. It is unclear whether the magnified view of the area at 8h corresponds to the area marked by the white rectangle. The structure within the rectangle appears lighter compared to the surrounding area, whereas the magnified view is dark, which creates confusion.

Furthermore, the lattice spacing should be expressed in angstroms.

Uranyl acetate (UA), used for negative staining, is known to induce precipitation of proteins and other soluble molecules. The authors incubated the condensates in UA for 5 minutes, which is unusually long. It would be more convincing to use a very short UA incubation and to apply chemical fixation prior to staining.

CLEM

The results presented in Figure 5A show that the fluorescence spots correspond to higher electron density regions in EM. The example appears to represent the longer time points (14h), as suggested by the presence of very large densities. However, according to the tomography data, the large aggregates are the only remaining discernible structures in the cell at this stage (aged condensates, Figure 5b, 14h), making them the default electron-dense structures. To strengthen the conclusions, CLEM at earlier stages of condensates (2h) would be highly beneficial. These images should be illustrated at higher magnification to enable the clear distinction between condensates, ribosome-rich areas, or other cellular structures. As demonstrated in Figure S14, protein condensates can exhibit two different aspects do these condensates share the same composition?

Lines 476-477 "However, aggregates within aged condensates were not observed, likely due to the resolution limitations of traditional two-dimensional (2D) electron microscopy (EM)". The presented EM images are of extremely low magnification, which limits the ability to assess finer structural details.

TEM tomography

The authors present tomographic images of two bacterial cells per time point (Figure 5b and Figure S14), but it is unclear how representative the observed target structures are. Figure S14 highlights this issue, showing that protein condensates can have two different structural aspects at the same time point. Are these condensates compositionally identical? CLEM analysis would be highly valuable to address this question.

The annotation of ribosomes in yellow (Figure 5 and Supplementary Movies) requires further justification. Many ribosome annotations overlap with nucleoid at the 2h and 8h time points. The ribosome annotations are particularly questionable at the 14H time point, where ribosomes appear totally absent in the cytoplasm. The wavy and irregular membrane morphology at 14H further supports that the cell may be dead or in a compromised state. This observation raises the possibility that the overexpression of the construct is toxic unless the authors can provide evidence to the contrary.

Reviewer #3

(Remarks to the Author)

In this study, Kang et al describe the time-dependent catalytic activity of biomolecular condensates prepared by constructs that harbor IDR as well as docking sites to recruit a fluorescent protein and an enzyme as reporter proteins. The researchers demonstrate that these condensates initially enhance reaction rates compared to reactions in solution, but their catalytic efficiency declines over time. This decline is associated with condensate aging, which they describe as transition from a liquid-like to a solid-like state. Both in vitro and in vivo experiments in *E. coli* confirm that condensate aging leads to protein aggregation and the loss of permeability, which affects enzymatic function by restricting access to active sites. The change of properties is analyzed in quantitative terms by a large set of methods: (i) Using fluorescence recovery after

photobleaching (FRAP) and particle tracking microrheology (PTM), the study reveals a progressive loss of molecular mobility as condensates age. (ii) Electron microscopy and tomography confirm that aged condensates accumulate protein aggregates. Finally, the study also demonstrates that small molecules like glycine can delay condensate aging and restore enzyme activity, while tryptophan accelerates aging and inhibits enzymatic function.

The teams used an exceptionally broad set of methods, particularly, several biophysical methods. FRAP to measure molecular mobility within condensates and reveals a decline over time, PTM to provides rheological insights, showing a transition from a liquid-like to a solid-like state. Cryo-EM and ET to directly visualize of protein aggregates in both synthetic and cellular condensates. Confocal Imaging to track enzyme localization and condensate morphology. Finally, CLEM was used to bridge insight gained at different scale. Since the condensates are designed to also recruit enzymes in the compartment, there is in general the chance to align biophysical insight with enzyme kinetic properties. In general, the study is technically sound, both in the use of the individual methods, as well as discussing data complementarily to achieve a well-founded description of aging.

Overall, the work by Kang et al. provides new unmatched insights into the properties of condensates, particularly by disclosing the role of aging processes of condensates and their description at the molecular level. There is surely high relevance of these data for biotechnological applications, both for employing the described system directly, but also for guiding analysis and tuning of other compartmentalization approaches. It is noted that it is difficult to estimate to which level the presented data are also directly relevant for condensates in natural processes, e.g., those that drive neurodegenerative diseases.

Based on the detailed insight the authors provide for describing a widely relevant biological process and the biotechnological relevance of this work, I strongly recommend the publication of the ms by Kang et al. in Nature Communications.

Minor and major points:

- (1) reference (2) is not a very suitable reference for fatty acid synthesis in megaenzymes, as it refers mainly the non-compartmentalized type II system.
- (2) On page 2, in the introduction section, the authors state „However, current studies predominantly based on the premise that biomolecular condensates remain in a liquid-like equilibrium state.“ . This statement should be supported by references.
- (3) Is there any difficulty arising from the fact that RIDD is dimeric? Does this interaction structure the condensate in any means?
- (4) What is the role of AEBSF in the buffer?
- (5) At some points, the authors refer to enzymes that have used, or „enzymes MenH-RIDD“. Is this a typo or have other enzymes than MenH been tested/pre-screened?
- (6) What is the molecular basis for the action of 1,6-hexanediol? Is it expected to have lipids incorporated in condensates?
- (7) A sentence to the preparation of samples for PTM in the Results section would support the understanding of this method. Similarly, the paragraph to PTM starting with „In this work, we focus...“ as well as the next one „Moreover, we studied...“ are difficult to understand for a broader biochemical community, and should be rephrased to improve readability. E.g. Flickian diffusion should be described in a sentence, or the reason for studying linear viscoelastic moduli should be explained.
- (8) Authors argue that „condensates resemble inclusion bodies“, based on the fact that 6M Guanidine hydrochloride is necessary to the dissolve the condensate. Is this really sufficient evidence for this argument? What does resemble mean exactly? Can the authors be more specific and add references?
- (9) The term and phenomenon „disorder-to-order transition“ should be explained more clearly. Is the beta-sheet structure a native fold or is this structure a sign of misfolding and self-assembly in beta sheet rich structures that is common to proteins. How should these structures nucleate aggregation - are there other examples, please add references.
- (10) Can the authors speculate on the molecular basis for condensates to lose the pseudo barrier upon aging. The argument that the reduced ability to exclude non-specific solutes is responsible for the loss of function seems very speculative. Why would this lead to reduced enzymatic efficiency. At an earlier stage in the ms, the loss in enzymatic efficiency was attributed to blocked access of active sites. How do these theories align?
- (11) Was arginine tested as a amino acid to aged condensates. It is often used in protein chemistry for proper folding of proteins due slightly amphiphilic properties.
- (12) In the discussion section, the statement „Despite the simplicity of this engineered system, it recapitulates key features of naturally occurring biomolecular condensates, including the selective enrichment of specific biomolecules and time-dependent changes in biophysical properties.“ is very speculative. Why is the system relevant for naturally occurring condensates.

Version 1:

Reviewer comments:

Reviewer #1

(Remarks to the Author)

The authors have addressed my previous concerns satisfactorily by new experiments and simulations, and I recommend this manuscript for publication. Minor comments are outlined below:

1. The added background on the passive rheology method is valuable, but the sentences beginning on line 322 reads too

informally for a scientific paper. Please revise the paragraph to a more formal academic style.

2. The previous comment on surface tension was to discuss surface tension in the context of compartmentalization (around line 700). The sentence added at line 182, while interesting, feels out of place in my opinion.

Reviewer #2

(Remarks to the Author)

I would like to thank the authors for their responses and for adding additional illustrations. I am fully satisfied with their replies to my comments 1–7 and comment 9.

I would just like to kindly point out typos in the following sentence: “The discrepancy in the diameters of biomolecular condensates at h_0 between TEM and confocal imaging is likely attributed to uranyl acetate staining during TEM preparation (Figure S11).” Additionally, Figure 3f was not updated to Ångström in the revised version of the manuscript.

I do not find the authors’ response to my comment 8 acceptable. The statement “Yellow dots only represent the non-condensate, ribosome-containing area, and do not correlate to any real cellular structures.” is problematic. Markers that do not correspond to any actual cellular structures lack scientific meaning and risk misleading the reader. To avoid confusion, the yellow dots should be removed from the figure.

Reviewer #3

(Remarks to the Author)

The manuscript has been thoroughly revised, and the newly added data provide valuable additional insights. The work is now well-suited for publication in Nature Communications.

The images or other third party material in this Peer Review File are included in the article’s Creative Commons license, unless indicated otherwise in a credit line to the material. If material is not included in the article’s Creative Commons license and your intended use is not permitted by statutory regulation or exceeds the permitted use, you will need to obtain permission directly from the copyright holder.

Responses to the Reviewers' Comments

We were pleased to receive constructive comments on the quality and significance of our work. In the revised version of the manuscript, we addressed all of the reviewers' comments point-by-point. These revisions have greatly enhanced the quality of the manuscript.

Comments to all Reviewers:

We thank all of the reviewers for their insightful comments on our manuscript. In the following section, we describe the specific changes/additions/corrections made to the manuscript based on reviewers' comments. Reviewer comments are listed first in *italic*, while our responses follow in plain text (color in **blue**), and changes/additions to the manuscript text are shown in as "quoted text" (color in **red**).

Response to Reviewer 1

We thank the reviewer for the insightful comments. The reviewer found "*This study is an important piece of information for understanding the connection between catalytic activity within condensates and their rheology, especially the aging process in vivo. I appreciate that the authors first conducted a significant study on in vitro condensates to understand the rheological transition during the aging process and then proceeded to a living E. coli system... I found the study to be well executed and a valuable contribution to the literature on condensate aging*".

Response to comments:

Comment 1:

The authors claim that the condensate ages through an intermediate Maxwell fluid phase: before this regime, the condensate is liquid-like, and after it, the condensate is solid-like. However, this claim is not entirely accurate because the frequency range probed by the particle tracking experiment is inherently limited by the experimental time scale and resolution. It is important to note that the reported material properties apply only to the specific experimental time scale or frequency range used. For example, there is no guarantee that an h0 condensate would not exhibit Maxwell fluid behavior at higher frequencies (i.e., on smaller time scales). I suggest probing a wider range of frequencies or, at the very least, explicitly acknowledging this limitation.

Response 1:

We really appreciate this comment and strongly agree with the reviewer on the limitation of particle tracking microrheology (PTM). In short, PTM relies on capturing the Brownian motion of particles to infer material properties like viscosity and elasticity. However, the speed at which particles move and the precision with

which we can track them are limited by the experimental setup. For example, at very short time-scales (i.e., high frequency), we might not be able to measure how particles move, affecting our understanding of high-frequency rheological behavior. At the same time, such limitations may result in missing particle positions, leading to inaccurate readings of material properties at high frequency. Thus, we agree with the reviewer that “reported material properties apply only to the specific experimental time scale or frequency range used.” However, PTM offers unique benefits for measuring the rheological properties that are hard to get from other methods, especially when studying small or complex materials such as LLPS, (and more importantly, LLPS is located inside of solvent environment), which is inaccessible to traditional macrorheology. To this end, we follow the reviewer’s suggestion to explicitly acknowledge this limitation in the revised manuscript.

Revisions to the manuscript:

Page 9 (line 28): We have revised the manuscript as follows: “Notably, the frequency range probed by the particle tracking experiment is inherently limited by the experimental time scale and resolution. Therefore, the reported material properties apply only to the specific experimental time scale or frequency range used.”

Comment 2:

The authors claim that restricted diffusivity alone is unlikely to affect the enzyme’s catalytic efficiency. However, the viscosity change shown in Figure S9 is quite minor. Studies such as Jawerth et al. (2020, Science) suggest that viscosity changes during aging can span several orders of magnitude. Therefore, I recommend investigating a wider range of viscosity changes and comparing them to the reaction rate.

Response 2:

We thank the reviewer for this valuable suggestion. And thanks for bringing up recent work (Louis Jawerth et al., *Science*, 370:1317-1323, 2020) on how viscosity affects diffusion. We further explored the role of viscosity in modulating enzymatic reactions, we expanded our investigation by testing a broader range of viscosities. This was achieved by incorporating varying amounts of sucrose into the reaction solutions (weight/weight). As shown in **Figure 1b**, the viscosity of the reaction solution increased with the sucrose concentration, generating a spectrum ranging from several mPa·s to several Pa·s.

Figure 1. (also Supplementary Figure S13 in the Supporting Information). Comparison of reaction rates among enzymes in sucrose solutions of varying viscosity. **(a)** Reaction rates of SHCHC (2-succinyl-6-hydroxy-2,4-cyclohexadiene-1-carboxylate) production were assessed in sucrose solutions at different concentrations. **(b)** Viscosity (η) measurements were conducted on these sucrose solutions using a rheometer. Data are expressed as the mean \pm s.d. ($n = 3$). Significance was calculated by unpaired t-test. ****: $P < 0.0001$; ***: $P < 0.001$; n.s (not significant): $P > 0.05$.

Notably, consistent with our previous findings, we observed no significant changes in reaction rates across this viscosity range (**Figure 1a**). These results suggest that, at least within the range of viscosity we investigated, increased viscosity alone does not appear to negatively impact the enzyme’s catalytic efficiency. However, we do acknowledge that restricted diffusion may still affect enzymatic reactions, particularly those involving large molecules. As mentioned in our manuscript: “while small-molecule reactions appear unaffected by restricted molecular diffusion under our experimental conditions, further investigation is required to assess the mobility of biomacromolecules (e.g., RNA) as substrates, where diffusion-limited effects may be more pronounced.”

Revisions to the manuscript:

Page 11 (line 44): We have revised the manuscript as follows: “Interestingly, we observed comparable reaction rates for the free enzyme in solution across varying viscosities achieved through different sucrose concentrations (**Figure S13**), which mimic the more viscous environment inside condensate during aging⁴³. Our result suggests that restricted diffusivity alone is unlikely to influence the catalytic efficiency of the enzyme, at least within the range of viscosities we investigated.”

Page 12 (line 2): We have added the following reference in the manuscript as Ref. 43:

Louis Jawerth et al. Protein condensates as aging Maxwell fluids. *Science*, **370**:1317-1323 (2020).

Revisions to the Supporting Information:

Page S23: We add Figure 1 as Figure S13 in the Supporting Information.

Comment 3:

*Since various mechanisms could be speculated regarding how *E. coli* might modulate condensate rheology, I recommend comparing the results to different types of modulation, such as altering salt concentration and pH, at least in vitro. It would be interesting to see if these environmental changes lead to similar effects on aging and catalytic activity.*

Response 3:

We appreciate this constructive suggestion. In response, we investigated the effects of salt concentration and pH on condensate aging and catalytic activity. To investigate the effects of salt concentration on condensates aging, we expressed and purified the phase-forming scaffold FIB1-GFP-RIAD and model fusion enzyme MenH-RIDD. A concentration of 10 μ M FIB1-GFP-RIAD was subsequently mixed with 1 μ M MenH-RIDD in a condensate formation buffer containing varying amounts of NaCl (50 mM, 150 mM and 250 mM). We further examined the morphological characteristics of condensates using confocal microscopy at 15 min (h0), 4 hours (h4), and 8 hours (h8) after mixing. Our findings indicate that condensates at lower salt concentrations transitioned to a solid-like state more rapidly than those at higher salt concentrations (**Figure 2**). Particularly, at 150 mM NaCl, the condensates gradually developed into larger clusters with non-spherical morphology, suggesting a liquid-to-solid transition (**Figure 2**). However, under the higher salt concentration of 250 mM NaCl, the condensates preserved their spherical shapes even after 8 hours, characteristic shapes of liquid-like condensates, indicating a delay of condensate aging (**Figure 2**). In contrast, at the salt concentration of 50 mM NaCl, irregular condensates were observed shortly after formation (h0), suggesting an acceleration of condensate aging (**Figure 2**). These results indicate that lower salt concentrations promote the aging of condensates.

Figure 2. (also Supplementary Figure S23 in the Supporting Information). Representative confocal images of catalytic condensates over time under different salt concentrations. Solutions containing 10 μM FIB1-GFP-RIAD and 1 μM MenH-RIDD were examined using confocal microscopy. Images were acquired at several time intervals. Scale bar, 20 μm .

We further performed fluorescence recovery after photobleaching (FRAP) experiments of catalytic condensates under different salt concentrations. At each time point following condensates formation, we observed a consistent trend in fluorescence recovery: lower salt concentrations resulted in decreased recovery (**Figure 3a**). This observation suggests that condensates under lower salt concentrations possess more solid-like characteristics. For example, at h0, FRAP revealed a fluorescence recovery of 39.7% for condensates at 50 mM NaCl, compared to 68.5% and 86.6% for condensates at 150mM NaCl and 250 mM NaCl, respectively. Consistently, the immobile fraction of the condensates decreased with salt concentration (**Figure 3b**). These findings confirm that condensate aging is accelerated at lower salt concentration.

Figure 3. (also Supplementary Figure S24 in the Supporting Information). FRAP analysis of catalytic condensates at h0, h4 and h8 under different NaCl concentrations. (a) Representative images of the condensates before, during, and after bleaching are shown (top). The bleached regions are indicated by red arrows. Scale bar, 10 μ m. The FRAP recovery curves of the catalytic condensates are presented (bottom). (b) Immobile fractions of the condensates derived from (a). Solutions containing 10 μ M FIB1-GFP-RIAD and 1 μ M MenH-RIDD were used for FRAP analysis. Data are represented as mean \pm s.d. (n = 5). All significance was calculated by unpaired t-test. ****: P < 0.0001; ***: P < 0.001; **: P < 0.01; *: P < 0.05; n.s (not significant): P > 0.05.

Moreover, we then investigated how the altered aging dynamics resulting from varying salt concentrations affect enzymatic reactions within the catalytic condensates. Three experimental groups were established: (1) 50 mM NaCl (catalytic condensates of MenH-RIDD in the reaction solution containing 50 mM NaCl), (2) 150 mM NaCl (catalytic condensates of MenH-RIDD in the reaction solution containing 150 mM NaCl) and (3) 250 mM NaCl (catalytic condensates of MenH-RIDD in the reaction solution containing 250 mM NaCl). To isolate the effect of condensate aging rather than a direct effect of altered salt concentrations on enzymatic activities, we compared the relative reaction rates, whereby the reaction rates of the systems with catalytic condensates under specific salt concentration was normalized against that of the control systems containing equivalent enzymes and identical salt concentration but lacking condensates. We found that lower salt concentration, which facilitate condensate aging, inhibits enzymatic reaction (**Figure 4**). This result aligns with our previous observation that the condensate aging promotor tryptophan inhibits the reaction, whereas condensate aging inhibitor glycine can partially rescue the impaired enzyme reaction.

Figure 4. (also Supplementary Figure S28 in the Supporting Information). Comparison of MenH-catalyzed reaction in catalytic condensates under different salt concentrations. Reaction rates for the catalytic condensates were normalized against a control system containing identical enzyme and NaCl concentration

but lacking condensates. The gray dashed line indicates a relative reaction rate of 1, representing a comparable rate between the system containing condensates and the control system. Data are represented as mean \pm s.d. ($n = 3$). All significance was calculated by unpaired t-test. ****: $P < 0.0001$; ***: $P < 0.001$; **: $P < 0.01$; *: $P < 0.05$; n.s (not significant): $P > 0.05$.

Extending this approach to pH modulation, we observed that higher pH values accelerated aging, evidenced by premature loss of spherical morphology (Figure 5) and reduced FRAP recovery (Figure 6). Conversely, lower pH delayed condensate aging (Figure 5 and Figure 6). Normalized activity measurements revealed that pH-dependent aging inversely correlated with catalytic efficiency (Figure 7).

Figure 5. (also Supplementary Figure S25 in the Supporting Information). Representative confocal images of catalytic condensates over time under different pH values. Solutions containing 10 μ M FIB1-GFP-RIAD and 1 μ M MenH-RIDD were examined using confocal microscopy. Scale bar, 20 μ m.

Figure 6 (also Supplementary Figure S26 in the Supporting Information). FRAP analysis of catalytic condensates at h0, h4 and h8 under different pH values. (a) Representative images of the condensates before,

during, and after bleaching are shown (top). The bleached regions are indicated by red arrows. Scale bar, 10 μm . The FRAP recovery curves of the catalytic condensates are presented (bottom). **(b)** Immobile fractions of the condensates derived from **(a)**. Solutions containing 10 μM FIB1-GFP-RIAD and 1 μM MenH-RIDD were examined using FRAP. Data are represented as mean \pm s.d. ($n = 4$). All significance was calculated by unpaired t-test. ****: $P < 0.0001$; ***: $P < 0.001$; **: $P < 0.01$; *: $P < 0.05$.

Figure 7. (also Supplementary Figure S29 in the Supporting Information). Comparison of relative reaction rates at different pH levels. Reaction rates for condensates were normalized to a control system containing identical enzyme and pH values but lacking condensates. ND (not detected): SHCHC absorbance signals were undetectable in h8 systems at pH 9.4, regardless of condensate presence. The gray dashed line indicates a relative reaction rate of 1, representing a comparable rate between the system containing condensates and the control system. Data are represented as mean \pm s.d. ($n = 3$). All significance was calculated by unpaired t-test. ****: $P < 0.001$; **: $P < 0.01$; *: $P < 0.05$.

These results collectively demonstrate that condensate aging is tunable through both small molecules and environmental parameters such as ionic strength and pH. However, we caution that these effects may be system-specific, as different phase-forming scaffold interaction networks could yield distinct environmental sensitivities.

Revisions to the manuscript:

Page 20 (line 42): We have added the following content to the manuscript: “Additionally, we explored whether condensate aging could be modulated by other factors such as salt concentration, pH, and arginine. We found that elevated salt concentrations, reduced pH values, or supplementation with arginine delay condensate aging (**Figure S23-S27**). Furthermore, we examined how salt concentration and pH influence reaction rates within condensates. Notably, we observed that impaired enzyme activities in aged condensates can be rescued by lower pH values and higher NaCl concentrations (**Figure S28 and S29**).

Together, these results indicate that multiple factors influence condensate aging, which can be leveraged to tune enzyme reaction rates within condensates. Further studies are required to investigate the generalizability of these factors in modulating the aging of other biomolecular condensates, as different phase-forming scaffold interaction networks could yield distinct environmental sensitivities.”

Revisions to the Supporting Information:

Page S34-S35: We have added Figure 2 and Figure 3 as Figure S23 and Figure S24 in the Supporting Information.

Page S36-S37: We have added Figure 5 and Figure 6 as Figure S25 and Figure S26 in the Supporting Information.

Page S39-S40: We have added Figure 4 and Figure 7 as Figure S28 and Figure S29 in the Supporting Information.

Comment 4:

It would be valuable to extend the experiments and discussion to explore how aging affects the conformations of both the scaffolds and the client enzyme, thereby impairing catalytic activity. Consulting with computational studies might help to assess the possible structural modification during the aging process: Garaizar et al. (PNAS, 2022), Takaki and Thirumalai (PNAS, 2024), Biswas and Potoyan (PTXLife, 2024), etc.

Response 4:

We thank the reviewer's valuable suggestion, and we agree with the reviewer that investigating the structural changes during the aging process with computational methods is highly valuable and will deepen our understanding of aging.

First, the three papers suggested by the review offer us with great insight into how aging affects the conformations of both the scaffolds and the client enzyme. A key mechanism by which aging affects protein conformations is through disorder-to-order structural transitions within the proteins themselves (Garaizar et al., *Proc. Natl. Acad. Sci. U.S.A.*, 119:e2119800119, 2022). For example, within the prion-like domains (PLDs) of proteins like FUS, segments known as low-complexity aromatic-rich kinked segments (LARKS) can undergo transitions to form kinked cross-beta sheets. These disorder-to-order transitions introduce physicochemical diversity and strengthen interprotein interactions among chemically identical proteins within the single-component condensate. In addition, the type of aggregated structure that forms during aging is influenced by factors like sequence complexity and monomer rigidity (Takaki and Thirumalai, *Proc. Natl. Acad. Sci. U.S.A.*, 121:e2409973121, 2024). Flexible monomers with low sequence complexity tend to form liquid-like droplets, while increasing monomer rigidity can lead to an abrupt transition to more

ordered structures resembling amyloid fibrils. Moreover, during aging solvent expulsion (desolvation) brings sticky regions closer together and promotes the transition from a viscous to an elastic state (Biswas and Potoyan, *PRX life*, 2:023011, 2024). The lifetime of "sticker" interactions between protein segments also profoundly influences the material properties, with long-lived, irreversibly cross-linked stickers leading to solid-like gels (Kelvin-Voigt model behavior) and shorter-lived stickers resulting in more fluid-like states (Maxwell fluid behavior). As suggested by the reviewer, we have added discussions on how aging affects the conformations of both the scaffolds and the client enzymes inspired by those simulation works.

More importantly, we ourselves conducted multiple-scale molecular dynamics (MD) simulations for both FIB1 scaffolds and the scaffold-enzyme (FIB1-GFP-RIAD/MenH-RIDD) systems. Five FIB1 intrinsically disordered regions (IDR) were used to study the aggregation and potential aging process of the scaffold with coarse-grained MD simulations. We found FIB1 IDRs scaffold initially formed aggregates after about 2.5 μs and maintained relative stability throughout the following 5 μs (**Figure 8a**). However, with the time traces, cluster dynamics showed progressive destabilization beyond the equilibrium phase, and partial dissociation dynamics became more frequent in the trajectory (**Figure 8b**). Especially around 7.5-8.5 μs , we observed a series of substantial structural reorganization events involving partial cluster disaggregation and dramatic conformational rearrangements, as evidenced by a sharp peak at 8.3 μs and smaller transient peaks emerging in the radius of gyration trajectory (**Figure 8a**), which collectively highlight the dynamic conformational alterations. This phenomenon may be associated with cluster aging processes. Subsequently, we performed all-atom md simulations to investigate whether the disaggregated IDR scaffold engaged in some unexpected interactions with the client enzyme that potentially affects the enzyme catalytic activity.

Figure 8. Coarse grained MD simulation results of five FIB1 IDRs. **(a)** The radius of gyration (Rog) curve of the system is shown, with a zoomed-in panel highlighting the 7.5–8.5 μs region. **(b)** Snapshots for coarse grained MD simulation.

The all-atom simulation system was built upon the trimeric RIAD/RIDD structural framework, where each RIAD domain interacts with two RIDD chains. Within this architecture, each RIDD chain could carry a client enzyme. Therefore, there are two MenH in the modeling system, which were designated as MenH1 and MenH2 for descriptive clarity. The model was generated using Chai-1, an AI-based protein modeling platform, followed by MD simulations with Amber24 and ff19SB force field. Through 500-ns MD simulations with 3 replicas, we observed the influence of FIB1 IDR affecting the activity of MenH1, which is much closer to the FIB1 IDR in space. By comparing the conformational dynamics of MenH1 and MenH2 in the modeling system with the native MenH-substrate complex system as a reference, distinct behavioral patterns were identified (**Figure 9b**). Notably, as part of MenH1's active site, the $\alpha 2'$ and $\alpha 3$ helices (aa: 146-170) underwent a remarkable shifting after 500 ns simulation (**Figure 10** and **Figure 11b**), which was induced by interactions with the FIB1 IDR. This shifting resulted in displacement of the substrate molecule within the MenH1 active pocket (as the distance curves in **Figure 9a**, and structure in **Figure 9c**) and significantly altered the binding free energy with the active site (average $\Delta G = -13.54 \pm 5.64$ kJ/mol, **Figure 11a**) compared to the native system (-41.18 ± 3.94 kJ/mol), while MenH2 surprisingly showed enhanced binding affinity ($\Delta G = -55.61 \pm 5.15$ kJ/mol), likely attributable to allosteric effects from GFP coupling.

Figure 9. (also Supplementary Figure S12 in the Supporting Information). All-atom MD simulation results for FIB1-GFP-RIAD/MenH-RIDD modeling system. (a) The curves of center of mass distance between His232 and substrate SEPHCHC (2-succinyl-5-enopyruvyl-6-hydroxy-3-cyclohexadiene-1-carboxylate) from different MenH-substrate systems. (b) Conformational snapshots from MD simulations of the FIB1-GFP-RIAD/MenH-RIDD model system, with MenH1 in cyan, MenH2 in blue, GFP in green, and FIB1 highlighted with a green outline. (c) A zoomed-in view of the MenH1 active site is shown, with a red dashed line indicating the distance between His232 and the substrate SEPHCHC.

Figure 10. Simulations results for MenH-substrate systems. Conformational alignments of the WT-MenH, MenH1, and MenH2 systems are displayed (top), where gray represents the initial structure and cyan denotes conformations sampled after 500 ns of simulations. Root-mean-square deviation (RMSD) profiles for MenH and the substrate SEPHCHC across these systems are shown (bottom).

Figure 11. Simulation results for MenH-substrate systems. (a) Comparative binding free energy analysis of MenH-substrate systems. (b) Root-mean-square fluctuation (RMSF) profiles across the three systems.

Revisions to the manuscript:

Page 11 (line 38): We have added the following: “It is also noteworthy that recent simulations reveal that aging significantly impacts protein conformations within biomolecular condensates⁵⁵⁻⁵⁷, such as disorder-to-order structural transitions and the formation of kinked cross-beta sheets. Desolvation could promote a transition from a viscous to an elastic state by bringing sticky regions closer. In addition, our simulation results also showed that aging could affect the enzyme catalytic activity through aggregation (**Figure S12**).”

Page 11 (line 40): We have added the following reference in the manuscript as Refs. 55-57:

1. Garaizar et al. Aging can transform single-component protein condensates into multiphase architectures. *Proc. Natl. Acad. Sci. U.S.A.*, **119**:e2119800119 (2022).
2. Takaki, R. and Thirumalai, D. Sequence complexity and monomer rigidity control the morphologies and aging dynamics of protein aggregates. *Proc. Natl. Acad. Sci. U.S.A.*, **121**:e2409973121 (2024).
3. Biswas, S. and Potoyan, D. A. Molecular drivers of aging in biomolecular condensates: desolvation, rigidification, and sticker lifetimes. *PRX life*, **2**:023011 (2024).

Comment 5:

One cannot determine a “solid-like network” structure solely from rheology experiments (e.g., as mentioned in lines 312 and 320). I recommend refraining from structural “network” references unless there is definitive evidence from the rheology data that supports such a claim. The same problem applies to the claim “weak gels” in line 332.

Response 5:

We thank the reviewer for raising this point. We delated the claim of “weak gels” and used the term “network” instead of “solid-like network” in the discussion of rheology experiments.

Revisions to the Supporting Information:

Page 10 (line 14 right): We have added the following: “By h8, α decreases significantly, approaching nearly 0, which suggests the formation of a network structure.”

Page 10 (line 36 right): We have deleted the following: “...demonstrating that the condensates are mechanically weak gels.”

Comment 6:

It would be valuable to include a discussion about the role of surface tension and how it relates to the ability of condensates to maintain isolated compartments.

Response 6:

We thank the reviewer for this valuable comment! Surface tension plays a significant role in nucleation, the process where a new phase begins to form within a parent phase, such as LLPS, which is commonly included in the calculation of total free energy change ΔG

$$\Delta G = \frac{4}{3}\pi r^3 \Delta G_v + 4\pi r^2 \gamma$$

Where γ is the surface tension and r is the radius of the new phase. Higher surface tension requires a larger critical nucleus size to overcome the energy barrier. In particular, surface tension affects the nucleation and growth of biological condensates in LLPS. For example, Martins’s group (Sárkány, Z. et al., *Adv. Sci.*, 10:2301501, 2023) recently discovered that surface tension effects (STEs) increase the critical solubility c_c compared to the saturation concentration c_∞ for small, curved nuclei, using the Ostwald-Freundlich equation

$$c_c = c_\infty \exp\left(\frac{\gamma v}{RTR_c}\right)$$

Where v is the molar volume of the protein and R is the universal gas constant. They developed a generalized nucleation-and-growth mode and found that the final size, as well as the moment of formation of biological condensates become highly regulated in the presence of strong STEs and that the evolution of droplet size distributions can be predicted based on c_c and c_∞ . Even though our current work focuses on the aging process, the surface tension effects on the kinetic process of phase separation will surely offer key insights in our future work.

Additionally, previous studies have demonstrated that surface tension influences the coexistence of multiple sub-phases within a single biomolecular condensate. Condensates formed through the liquid-liquid phase separation of intrinsically disordered proteins can exhibit various internal sub-phases that are immiscible with one another. Examples of hierarchically organized condensates include nucleolus (Feric et al., *Cell*, 165:1686-1697, 2016), paraspeckles (West et al., *J. Cell Biol.*, 214:817-830, 2016), and stress granules (Jain et al., *Cell* 164:487-498, 2016). For example, the nucleolus comprises at least three sub-compartments: the fibrillar center (FC), which houses the RNA polymerase machinery; the dense fibrillar component (DFC), enriched in the protein fibrillarin (FIB1); and the granular component (GC) enriched in the protein nucleophosmin (NPM1). It has been found that sub-phases with relatively high surface tension (more hydrophobic) tend to be enveloped by sub-phases with relatively low surface tension (more hydrophilic). Particularly, due to the greater hydrophobicity of FIB1 compared to NPM1, with $\gamma_{\text{FIB1/water}} > \gamma_{\text{NPM1/water}}$, FIB1 droplet will tend to be enveloped by NPM1 droplet within the nucleolus (Feric et al., *Cell*, 165:1686-1697, 2016). Together, these findings highlight the potential to modulate the formation and organization of condensates through the use of surfactants. Indeed, recent work by Miriam Linsenmeier et al. indicates that surfactant molecules can delay condensate aging (Miriam Linsenmeier et al. *Nat. Chem.*, 15:1340-1349, 2023).

Revisions to the manuscript:

Page 6 (line 14): “It is noteworthy that the size distribution and the maintenance of sub-phases of the condensates may be regulated in the presence of surface tension effects ^{3,45}.”

Page 6 (line 16): We have added the following reference in the manuscript as Ref. 3 and Ref. 45:

1. Feric, M. et al. Coexisting liquid phases underlie nucleolar subcompartments. *Cell*, **165**:1686-1697 (2016).
2. Sárkány, Z. et al. Quantification of surface tension effects and nucleation-and-growth rates during self-assembly of biological condensates. *Adv. Sci.*, **10**:2301501 (2023).

Response to Reviewer 2

We really appreciate the thoughtful and constructive feedback on our electron microscopy (EM) analysis. As suggested, we conducted additional experiments to address the points raised. We believe these improvements significantly enhance the technical depth and reproducibility of our analysis.

Response to comments:

Comment 1:

Negative staining was used to image in vitro assembled condensates, as illustrated in Figure 3f. This image appears intended to provide higher-resolution views of the protein aggregates shown in Figure 3a. However, I find it difficult to correlate the sizes of condensates between Figure 3a and Figure 3f. The scale bar in Figure 3f (0h) corresponds to 200 nm, and the condensate shown is approximately 300 nm in diameter. In contrast, judging from the scale bar in Figure 3a (10 microns) and the data in Figure S14, where the average condensate diameter is reported as 4 microns. To clarify this discrepancy, I suggest including a series of low-magnification EM images displaying multiple condensates for each condition (0h, 4h, 8h) in the Supplementary Materials.

Response 1:

We appreciate the reviewer's question. We imaged the condensate at low magnification using transmission electron microscopy (TEM), confirming that condensates at h0 displayed smaller diameters compared to confocal imaging (**Figure 12**). A similar discrepancy regarding the sizes of biomolecular condensates measured by different imaging techniques has been documented in previous studies (Ray et al., *Nat. Chem.*, 12:705-716, 2020; Amit Netzer et al., *Proc. Natl. Acad. Sci.*, 120: e2310569120, 2023; Xi Li et al., *Nat. Commun.*, 15: 8748, 2024). In these studies, confocal imaging revealed biomolecular condensate diameters in the several-micron range, while TEM indicated diameters on the order of several hundred nanometers. In confocal imaging, we visualize the condensates through fluorescent tags directly, whereas TEM imaging relies on staining with uranyl acetate (UA).

Figure 12. (also Supplementary Figure S9 in the Supporting Information). TEM imaging of h0, h4 and h8 biomolecular condensates at low magnification. Scale bar, 2 μm .

We postulate that the apparent reduction in size observed with TEM is likely due to UA, which is an acidic solution that may influence the formation of biomolecular condensates, leading to decreased diameters. To test this hypothesis, we imaged the solution containing biomolecular condensates at various points after formation, both with or without the treatment of UA, using confocal microscopy. As shown in **Figure 12**, the biomolecular condensates at h0 exhibited diameters in the range of several micrometers without UA treatment. In contrast, following UA treatment, the h0 biomolecular condensates displayed decreased diameters, measuring in the range of several hundred nanometers (**Figure 13**), indicating that the liquid-like condensates at h0 are more susceptible to UA. Notably, the condensates at h4 and h8 exhibited similar sizes regardless of UA treatment (**Figure 13**), reflecting their more solid-like state.

Figure 13. (also Supplementary Figure S11 in the Supporting Information). Confocal imaging of biomolecular condensates with or without 2% (w/v) uranyl acetate (UA) treatment. Representative confocal images of biomolecular condensates at different stages are shown. Condensates at h0 exhibited decreased sizes, whereas those at h4 and h8 showed comparable size after UA treatment, suggesting that the liquid-like biomolecular condensate at h0 are more susceptible to UA. Scale bar, 20 μm .

Together, these data suggest that the observed discrepancy in condensate diameters between confocal and TEM imaging likely arises from the effect of UA staining during sample preparation for TEM analysis.

Revisions to the Manuscript:

Page 11 (line 21): We have added the following content to the manuscript: “The discrepancy in the diameters of biomolecular condensates at h0 between TEM and confocal imaging is likely attributed to uranyl acetate staining during TEM preparation (**Figure S11**).”

Revisions to the Supporting Information:

Page S19: We have added Figure 12 as Figure S9 in the Supporting Information.

Page S21: We have added Figure 13 as Figure S11 in the Supporting Information.

Comment 2:

Additionally, the field of view of the magnified insets in Figure 3f is too small to reliably correlate the indicated features with those in the upper-row images. It is unclear whether the magnified view of the area at h8 corresponds to the area marked by the white rectangle. The structure within the rectangle appears lighter compared to the surrounding area, whereas the magnified view is dark, which creates confusion.

Response 2:

We thank the reviewer for the constructive suggestion. The magnified view of the area at h8 condensates indeed belongs to the upper image at lower magnification. The lighter appearance of structures in the lower magnification images likely results from an averaging effect, where lighter and darker areas blend together due to the larger field of view being captured. To establish a clearer relationship between the structures in the magnified images and those at lower magnification, we have imaged the condensates at various magnifications (**Figure 14**).

Figure 14. (also Supplementary Figure S10 in the Supporting Information). TEM images of catalytic condensates at h0, h4 and h8 at a series of magnifications. Colored boxes denote different magnifications. The white dashed lines outline the edges of the condensates. Protein aggregates are marked by magenta arrowheads. Scale bars, 200 nm (black boxes), 50 nm (yellow boxes), 20 nm (blue boxes), 10 nm (green boxes), 5 nm (red boxes) and 1 nm (cyan boxes).

Revisions to the Supporting Information:

Page S20: We have added Figure 14 as Figure S10 in the Supporting Information

Comment 3:

The lattice spacing should be expressed in angstroms.

Response 3:

We appreciate the reviewer's suggestion. The lattice spacing is now expressed in angstroms (**Figure 15**)

Figure 15. (also Figure 3f in the manuscript). TEM images of catalytic condensates at different stages. Particles with increased electron density are observed over time, indicating the formation of protein aggregates (top). Higher magnification images reveal a disorder-to-order transition, with an increase in crystallized area exhibiting an approximately 3.5 Å lattice spacing characteristic of β -sheet structures (bottom). Protein aggregates are indicated by red arrowheads. The white dashed lines outline the edge of the condensates. Scale bars, 200 nm (top) and 0.5 nm (bottom).

Revisions to the manuscript:

Page 13 (line 2): We have added Figure 15 as Figure 3f and redcribed the lattice spacing in angstroms in the manuscript.

Comment 4:

Uranyl acetate (UA), used for negative staining, is known to induce precipitation of proteins and other soluble molecules. The authors incubated the condensates in UA for 5 minutes, which is unusually long. It would be more convincing to use a very short UA incubation and to apply chemical fixation prior to staining.

Response 4:

We agree with the reviewer that, in traditional negative staining protocols, samples are typically incubated with 1-2% (w/v) UA solutions for 30 seconds to 1 minute. However, for samples containing protein condensates, previous studies have also employed longer incubation times of 3–5 minutes (Childers et al., *Langmuir*, 28:6386-6395, 2012; Abeyasinghe et al., *Small*, 20:2308390, 2024, Poudyal et al., *Nat. Commun.*, 14:6199, 2023). We postulate that the extended incubation time for samples containing biomolecular condensates is likely due to their distinct microenvironments, such as pH and polarity (Dai et al., *Cell*, 187:1-16, 2024; Küffner et al., *ChemSystemsChem*, 2:e2000001, 2020), which may influence UA's interactions with target proteins. While we did test shorter incubation durations, periods less than 5 minutes did not yield satisfactory contrast. Therefore, we opted to incubate the samples containing condensates in UA for 5 minutes in the subsequent assays.

Comment 5:

The results presented in Figure 5A show that the fluorescence spots correspond to higher electron density regions in EM. The example appears to represent the longer time points (14h), as suggested by the presence of very large densities. However, according to the tomography data, the large aggregates are the only remaining discernible structures in the cell at this stage (aged condensates, Figure 5b, 14h), making them the default electron-dense structures. To strengthen the conclusions, CLEM at earlier stages of condensates (2h) would be highly beneficial. These images should be illustrated at higher magnification to enable the clear distinction between condensates, ribosome-rich areas, or other cellular structures.

Response 5:

We thank the reviewer for this insightful and constructive comment. As suggested, we performed correlative light-electron microscopy (CLEM) on *E. coli* cells expressing synthetic condensates (Erluc1) at earlier time points. These new data, presented in **Figure 16** and **Figure 17**, include higher-magnification images to enable clearer differentiation between condensates, ribosome-rich regions, and other cellular structures.

Figure 16. (also Figure 5a in the manuscript) CLEM of *E. coli* strain expressing synthetic condensates (Erluc1). Representative confocal image overviews are shown (upper left), with the correlated regions further visualized by EM (bottom left). Regions highlighted in yellow and orange boxes are magnified in the right panels. Cells were fixed using high-pressure freezing and embedded in resins. Thin-sections (100 nm) were incubated with an anti-GFP primary antibody followed by a fluorescently tagged secondary antibody. Confocal microscopy images were aligned with TEM images based on cell structures and distribution. Scale bars, 10 μm (confocal) and 4 μm (TEM).

Figure 17. (also Supplementary Figure S18a in the Supporting Information) CLEM analysis of Erluc1 at 14 h. Imaging and processing protocols match those described for Figure 16. Scale bars, 10 μm (confocal) and 4 μm (TEM).

Revisions to the Manuscript:

Page 16 (line 1): We have added Figure 16 as Figure 5a in the manuscript.

Revisions to the Supporting Information:

Page S28-S29: We have added Figure 17 as Figure S18a in the Supporting Information.

Comment 6:

Lines 476-477 “However, aggregates within aged condensates were not observed, likely due to the resolution limitations of traditional two-dimensional (2D) electron microscopy (EM)”. The presented EM images are of extremely low magnification, which limits the ability to assess finer structural details.

Response 6:

We thank the reviewer for this insightful suggestion. The EM images from the CLEM analysis are now presented at higher magnification (see details in Response 5 and Response 7). Accordingly, we have removed the aforementioned sentence.

Revisions to the Manuscript:

Page 15 (line 11 right): The following sentence has been deleted from the manuscript: “**However, aggregates within aged condensates were not observed, likely due to the resolution limitations of traditional two-dimensional (2D) electron microscopy (EM).**”

Comment 7:

The authors present tomographic images of two bacterial cells per time point (Figure 5b and Figure S14), but it is unclear how representative the observed target structures are. Figure S14 highlights this issue, showing that protein condensates can have two different structural aspects at the same time point. Are these condensates compositionally identical? CLEM analysis would be highly valuable to address this question.

Response 7:

We thank the reviewer for the comment. As noted in our revised analysis (Figure 16 and Figure 17), CLEM imaging of *E. coli* at earlier post-induction time points confirms that fluorescent signals colocalize with electron-dense regions, supporting their identity as biomolecular condensates formed by phase-separating scaffolds. The structural heterogeneity observed in condensates at a single time point (Figure 18) likely reflects differences in maturation states. Specifically, variations in protein concentration or maturation time may lead to differences in electron density or architecture during condensate assembly. To provide more evidence of our condensate formation process from different time points, we also added a TEM images gallery from 2 h to 14 h in Figure 18.

Figure 18. (also Supplementary Figure S18c in the Supporting Information). TEM gallery of *E. coli* expressing synthetic condensates at 2 h, 4 h, 8 h and 14 h post-induction. Scale bars, 500 nm.

Revisions to the Supporting Information:

Page S28-29: We have added Figure 18 as Figure S18c in the Supporting Information.

Comment 8:

The annotation of ribosomes in yellow (Figure 5 and Supplementary Movies) requires further justification. Many ribosome annotations overlap with nucleoid at the 2h and 8h time points. The ribosome annotations are particularly questionable at the 14H time point, where ribosomes appear totally absent in the cytoplasm.

Response 8:

We thank the reviewer for identifying this annotation discrepancy. The yellow spheres in **Figure 5c** and **Supplementary Movies** were intended to denote ribosome-rich regions rather than individual ribosome locations, as erroneously described in the original manuscript. We have corrected this error in the manuscript and redescribed the 3D model images in **Figure 5c**.

Revisions to the Manuscript:

Page 17 (line 22): We have revised the description of the “yellow dots” in Figure 5c as follows: “yellow dots only represent the non-condensate, ribosome containing area, and do not correlate to any real cellular structures.”

Comment 9:

The wavy and irregular membrane morphology at 14H further supports that the cell may be dead or in a compromised state. This observation raises the possibility that the overexpression of the construct is toxic unless the authors can provide evidence to the contrary.

Response 9:

We thank the reviewer for raising this important concern. The irregular membrane morphology observed at 14 h is unlikely to reflect cell death or toxicity caused by condensate overexpression. Instead, we would attribute these features to suboptimal fixation of late-stage bacterial cultures, as prolonged incubation times inherently challenge sample preservation. Supporting this, **Figure 19** demonstrates that *E. coli* strains not expressing condensates (Erluc0) exhibit similar membrane irregularities when subjected to extended incubation and high-pressure freezing.

Figure 19. TEM images of Erluc0 fixed via high-pressure freezing after extended incubation. Scale bars, 1 μm .

Additionally, to investigate whether the overexpression of the biomolecular condensates is toxic to the *E. coli* cells, we monitored the cell growth rates by measuring OD_{600} for the Erluc1 expressing both RIDD-tagged luciferase (Rluc-RIDD) and the phase-forming scaffold FIBI-GFP-RIAD. As a control, we utilized the Erluc0 strain, which expresses only the enzyme Rluc-RIDD. As shown in **Figure 20**, we observed similar cell growth rates between the two strains. This finding suggests that the expression of the biomolecular condensate is not likely to impose cell toxicity to the *E. coli*.

Figure 20. Comparison of bacterial growth rates by monitoring OD_{600} between Erluc1 and Erluc0. The overexpression of condensates had no significant effect on the growth of engineered strains. Data are expressed as mean \pm s.d. ($n = 3$). t_i represents the time post-induction of condensate phase-forming scaffolds (FIB1-GFP-RIAD).

Together, these results demonstrate that the expression of the biomolecular condensates does likely result in a compromised state of the cells; rather, the wavy and irregular membrane morphology of cells at the late stage is likely attributed to factors such as nutrient depletion and modifications in membrane fluidity caused by prolonged cultivation times.

Response to Reviewer 3

We thank the reviewer for the insightful comments. The reviewer found “*The teams used an exceptionally broad set of methods, particularly, several biophysical methods... In general, the study is technically sound, both in the use of the individual methods, as well as discussing data complementarily to achieve a well-founded description of aging...Based on the detailed insight the authors provide for describing a widely relevant biological process and the biotechnological relevance of this work, I strongly recommend the publication of the ms by Kang et al. in Nature Communications.*”

Response to comments:

Comment 1:

Reference (2) is not a very suitable reference for fatty acid synthesis in megaenzymes, as it refers mainly the non-compartmentalized type II system.

Response 1:

We appreciate the reviewer's insightful comment regarding the reference appropriateness. The reviewer is absolutely correct that reference 2, entitled “Enzymology of assembly line synthesis by modular polyketide synthases”, primarily discusses modular polyketide synthases, making it less suitable as a reference for fatty acid synthesis. To address this concern, we have: (1) Swapped the positions of "polyketide synthases" and "fatty acid synthases" in the sentence to better align with the respective references. (2) Ensured that reference 1, which describes fatty acid synthase as a prototypical multienzyme complex, now properly supports the mention of fatty acid synthases.

We have implemented these changes in the manuscript, ensuring proper alignment between the text and references. The revised text now reads: **the formation of multi-enzyme complexes, such as fatty acid synthases and polyketide synthases, is crucial for enhancing and regulating enzymatic flux^{1,2}.** The modified sentence now accurately reflects the appropriate reference for each enzyme system.

Revisions to the manuscript:

Page 3 (line 4): We have revised the manuscript as follows: **“the formation of multi-enzyme complexes, such as fatty acid synthases and polyketide synthases, is crucial for enhancing and regulating enzymatic flux^{1,2}.”**

Comment 2:

On page 2, in the introduction section, the authors state „However, current studies predominantly based on the premise that biomolecular condensates remain in a liquid-like equilibrium state.“ . This statement should be supported by references.

Response 2:

We thank the reviewer for this suggestion. We have added references to support our statement. The cited reviews discuss the influence of liquid-liquid phase separation on enzymatic catalysis, yet the studies therein primarily assume that biomolecular condensates maintain equilibrium without accounting for potential temporal changes in their biophysical properties.

Revisions to the manuscript:

Page 3 (line 21): The following references have been added as Ref. 23 and Ref. 24:

1. Samuel Lim and Douglas S. Clark. Phase-separated biomolecular condensates for biocatalysis. *Trends in Biotechnol.*, **42**:496-509 (2024).
2. Brian G. O'Flynn and Tanja Mittag. The role of liquid–liquid phase separation in regulating enzyme activity. *Curr. Opin. in Cell Biol.*, **69**:70-79 (2021).

Comment 3:

Is there any difficulty arising from the fact that RIDD is dimeric? Does this interaction structure the condensate in any means?

Response 3:

We appreciate the reviewer's insightful question. As noted, RIDD is indeed a dimeric protein capable of binding its complementary peptide RIAD to form a stable trimer, as characterized in our prior study (Wei Kang et al., *Nat. Commun.*, 10:4248, 2019). The RIDD-RIAD system offers distinct advantages, including nanomolar binding affinity and small molecular sizes, enabling minimum interference with the folding and minimal steric interference with fused proteins, making it well-suited for recruiting RIDD-tagged proteins into RIAD-decorated condensates. Importantly, we encountered no challenges in expressing or purifying RIDD-fused proteins.

In respect of the role of RIDD's dimeric state in condensate formation, we posit that this multivalency is likely to facilitate phase separation. While intrinsically disordered regions (IDRs) are known to drive condensate formation through weak, transient interactions, multivalent interactions between folded domains also play a critical role in condensate formation. For example, the oligomerization domain of nucleophosmin (NPM1), essential for pentamer formation, is required for formation of NPM1 condensates

(Feric et al. *Cell*, 165:1686-1697, 2016; Mitrea D.M. et al., *Proc. Natl. Acad. Sci.*, 111:4466-4471, 2014). Additionally, Shin et al. fused a light-induced protein oligomerization domain to IDR, generating a synthetic protein that formed biomolecular condensates upon light stimulation (Shin et al., *Cell*, 168:159-171, 2017). These data indicate that the combination of specific oligomerization domains with IDRs is indeed a potent mechanism to drive condensate formation. Together, these examples underscore how multivalency among folded proteins can facilitate condensate formation. Thus, RIDD's dimeric architecture plausibly enhances condensate formation, though further studies are needed to quantify the contribution of these interactions to this process.

Comment 4:

What is the role of AEBSF in the buffer?

Response 4:

AEBSF (4-(2-aminoethyl) benzenesulfonyl fluoride hydrochloride) was included in the buffer to prevent protein degradation. In our study, we observed that FIB1 is particularly susceptible to proteolysis upon cell lysis (as shown in **Figure 21**). To mitigate this issue, we supplemented the buffer with 0.5 mM AEBSF, a serine protease inhibitor that covalently modifies the active center of protease such as proteinase K, trypsin, and plasmin (Narayanan and Jones, *Chem. Sci.*, 6:2650-2659, 2015). Moreover, AEBSF is considered as a less toxic alternative to the commonly used protease inhibitor PMSF (phenylmethylsulfonyl fluoride).

Figure 21. SDS-PAGE analysis of FIB1 fusion protein stability. Comparison of samples prepared with or without AEBSF demonstrates the inhibitor's efficacy in preventing proteolytic degradation. FIB1-GFP-RIAD (44.4 kDa).

Comment 5:

At some points, the authors refer to enzymes that have used, or „enzymes MenH-RIDD“. Is this a typo or have other enzymes than MenH been tested/pre-screened?

Response 5:

We thank the reviewer for identifying this error. This was indeed a typographical mistake, and we have corrected it in the revised manuscript.

Revisions to the manuscript:

Page 7 (line 25): The sentence has been corrected to read: “FRAP analysis of MenH-RIDD demonstrated a similar decrease in molecular mobility over time.”

Comment 6:

What is the molecular basis for the action of 1,6-hexanediol? Is it expected to have lipids incorporated in condensates?

Response 6:

The rationale for using hexanediol to disrupt phase-separated condensates stems from a pioneering study, in which 1,2-hexanediol and cyclohexane-1,2-diol were shown to perturb the permeability of the nuclear pore complex, a phase-separated multiprotein assembly driven by hydrophobic interactions between phenylalanine/glycine repeat domains (Ribbeck and Görlich, *The EMBO J.*, 21:2664-2671, 2002). Subsequent work on nuclear transport mechanisms tested different alcohols and identified 1,6-hexanediol as a potent inducer of nuclear pore permeability and demonstrated that the efficacy of aliphatic alcohols in dissolving condensates correlates with their hydrophobicity (Shulga and Goldfarb, *Mol. Cell Biol.*, 23:534-542, 2003). Since then, owing to the sensitivity of the biomolecular condensates to alkanediols, 1,6-hexanediol has been widely employed as hydrophobic disruptors to probe the biophysical properties of the phase-separated condensates both in vitro and in cells (Hang sun et al, *Nat. Nanotechnol.*, 19:1354-1375, 2024; Harper et al., *Nat. Commun.*, 15:10240, 2024; Guo et al. *Science*, 385:eadf4478, 2024).

However, as the reviewer notes, the molecular mechanism underlying the disruptive effects of alkanediols remain elusive. While they are often regarded as hydrophobic disruptors, this simplest view fails to fully account for their efficacy, given that biomolecular condensates are stabilized by diverse interactions, including cation- π , electrostatic, hydrogen bonding interactions in addition to hydrophobic interaction (Boeynaems S. et al., *Trends Cell Biol.*, 28:420-435, 2018). Ribbeck and Görlich proposed that alkanediols act as ‘mini detergent’, with an ethylene glycol polar group and an aliphatic apolar moiety (Ribbeck and Görlich, *The EMBO J.*, 21:2664-2671, 2002). More recently, all-atom simulations by Zheng et al. revealed that the aliphatic segment of 1,6-hexanediol interacts with hydrophobic and aromatic amino acids such as tyrosine, while its polar hydroxyl groups form hydrogen bonds with polar residues such as glutamine, suggesting it may compete with these types of residue for protein-protein contacts critical for phase separation (Zheng et al., *The EMBO J.*, 1-16, 2025).

Given the physiological and pathological significance of biomolecular condensates formed via LLPS, further studies are warranted to elucidate the precise molecular basis of alkanediol-mediated disruption and to develop more specific and potent small-molecule modulators of condensate dynamics. However, such investigations lie beyond the scope of the present work.

Comment 7:

A sentence to the preparation of samples for PTM in the Results section would support the understanding of this method. Similarly, the paragraph to PTM starting with „In this work, we focus...“ as well as the next one „Moreover, we studied...“ are difficult to understand for a broader biochemical community, and should be rephrased to improve readability. E.g. Flickian diffusion should be described in a sentence, or the reason for studying linear viscoelastic moduli should be explained.

Response 7:

We thank the reviewer for this comment. We have added many more details about particle tracking microrheology (PTM) in the result section so that a broader biochemical community could have better readability. Changes can be found below.

Revisions to the manuscript:

Page 9 (line 22): We have added the following: “In short, PTM is a passive microrheological technique that measures the rheological properties of materials by tracking the Brownian motion of embedded microscopic particles.”

Page 10 (line 6): We have added the following: “The diffusive exponent $\alpha(\tau)$ indicates how fast these particles spread out over time, can change at short times due to local quirks. But at long times, $\alpha(\tau)$ settles into a constant value. In this work, we focus on the long-time limit of the diffusion exponent. such that the diffusive exponent $\alpha(\tau) = \alpha$ is constant.”

Page 10 (line 12): We have added the following: “Normally, particles spread out with a diffusive exponent $\alpha = 1$, called Fickian diffusion. At the beginning (t_0), probe particles exhibit diffusive exponent $\alpha = 0.8054$, close to Fickian diffusion.”

Page 10 (line 17): We have added the following: “The storage modulus G' characterizes the elastic response of the droplets, showing how much energy the material can store and bounce back. It’s about the material’s stiffness, acting like a solid. The loss modulus G'' characterizes the viscous component of the response, showing how much energy the material loses as heat. It’s about the material’s flow, acting like a liquid.”

Comment 8:

Authors argue that „condensates resemble inclusion bodies“, based on the fact that 6M Guanidine hydrochloride is necessary to dissolve the condensate. Is this really sufficient evidence for this argument? What does resemble mean exactly? Can the authors be more specific and add references?

Response 8:

Thanks for this constructive suggestion. Our initial comparison between aggregated species in aged condensates and inclusion bodies (IB) was based on their shared insolubility in standard buffers and their solubilization upon treatment with 6M guanidine hydrochloride. IBs, compartments of insoluble, misfolded protein are well-documented in recombinant protein expression. However, we fully agree with the reviewer that solubility behavior alone is insufficient to support the claim that aged condensates "resemble" inclusion bodies. Consequently, we have removed this argument from the manuscript.

Revisions to the manuscript:

Page 11 (line 14): We have deleted the following statement: “These findings suggest that the aggregated species in aged condensates resemble inclusion bodies formed by partially and/or misfolded proteins, which necessitate denaturation for solubilization.”

Comment 9:

The term and phenomenon „disorder-to-order transition“ should be explained more clearly. Is the beta-sheet structure a native fold or is this structure a sign of misfolding and self-assembly in beta sheet rich structures that is common to proteins. How should these structures nucleate aggregation - are there other examples, please add references.

Response 9:

Thanks for this comment. We apologize for the confusion. What we meant in the manuscript about the “disorder-to-order transition” was that there was more crystalline area as shown in the TEM. To avoid the confusion, we have deleted the term “disorder-to-order transition” in the revised manuscript.

Revisions to the manuscript:

Page 11 (line 27): We have revised the manuscript as follows: “Higher-resolution TEM images revealed an increase in short-range ordered crystalline structures with a lattice spacing of approximately 3.5 Å (**Figure 3f**), ...”

Comment 10:

Can the authors speculate on the molecular basis for condensates to lose the pseudo barrier upon aging. The argument that the reduced ability to exclude non-specific solutes is responsible for the loss of function seems very speculative. Why would this lead to reduced enzymatic efficiency. At an earlier stage in the ms, the loss in enzymatic efficiency was attributed to blocked access of active sites. How do these theories align?

Response 10:

Previous studies have demonstrated that condensates formed by different biomolecules have distinct chemical environments that dictate interactions between client molecules and these scaffolds, leading to either attraction or repulsion of solutes by the condensates (Kilgore et al, *Nat. Chem. Biol.*, 18:1298-1306, 2022). For example, certain anti-cancer drugs have been observed to concentrate within specific biomolecular condensates through interactions with the proteinaceous environment rather than directly targeting their intended substrates (Klein et al, *Science*, 368:1386-1392, 2020). For our study, we hypothesize that the chemical solvating properties of the condensates change over time, leading to altered permeability. Additionally, previous studies have shown that the internal environments of condensates can significantly influence biomolecular activity (Kilgore et al, *Nat. Chem. Biol.*, 20:291-301, 2024). Further studies are warranted to investigate how the altered permeability of the condensates influences enzymatic activities.

Revisions to the manuscript:

Page 18 (line 9): We have revised the manuscript as follows: "... indicating that their ability to exclude non-specific solutes was compromised, likely due to the altered chemical solvating properties of the condensates over time⁶⁰. This loss of the pseudo barrier underscores the failure of aged condensates to function as membraneless organelles, as they can no longer maintain a compartmentalized environment. Given that the internal environment of condensates can influence biomolecular activities⁶¹, further investigation is necessary to explore how the altered permeability of condensates affects enzyme activities within these structures."

Page 18 (line 10): We have added the following reference in the manuscript as Ref. 60:

Henry R. Kilgore and Richard A. Young. Learning the chemical grammar of biomolecular condensates. *Nat. Chem. Biol.*, **18**:1298–1306 (2022).

Page 18 (line 13): We have added the following reference in the manuscript as Ref. 61:

Kilgore et al. Distinct chemical environments in biomolecular condensates. *Nat. Chem. Biol.*, **20**:291–301 (2024).

Comment 11:

Was arginine tested as a amino acid to aged condensates. It is often used in protein chemistry for proper folding of proteins due slightly amphiphilic properties.

Response 11:

We appreciate the reviewer's insightful suggestion. We investigated the potential of arginine as a modulator of condensate aging. To this end, we prepared biomolecular condensates by mixing 6 μM FIB1-GFP-RIAD and 1 μM MenH-RIDD. Then, we analyzed molecular motion within these condensates in the presence or absence of 0.2 mM arginine using FRAP analysis at several time intervals. We expected that if arginine inhibits condensate aging, biomolecules within the condensate would exhibit increased diffusibility. Our findings revealed that fluorescence recovery was significantly higher in the presence of arginine compared to conditions without it at all time points (**Figure 22a**), indicating an inhibition of condensate aging. Correspondingly, the immobile fraction decreased upon the addition of arginine (**Figure 22b**). Collectively, these results demonstrate that arginine can act as an inhibitor of condensate aging under our experimental conditions. However, we contend that further studies are necessary to assess the generalizability of this modulator, as different condensates may respond variably to arginine.

Figure 22. (also Supplementary Figure S27 in the Supporting Information) Arginine inhibits condensate aging. (a) FRAP analysis of biomolecular condensates with (+Arg) or without (-Arg) supplementation of 0.2 mM arginine. Representative images of condensates at various stages following formation, before, at, and after bleaching, are shown (top). The bleached regions are indicated by white arrows. Scale bar, 2 μm . FRAP recovery curves for h0, h4, and h8 condensates are displayed (bottom). (b) Immobile fractions of condensates with (+Arg) or without (-Arg) arginine at varying ages derived from panel (a). Data are expressed as mean \pm s.d. ($n = 4$). Significance was calculated by unpaired t-test. **: $P < 0.01$; *: $P < 0.05$.

Revisions to the manuscript:

Page 20 (line 42): We have added the following content to the manuscript: “Additionally, we explored whether condensate aging could be modulated by other factors such as salt concentration, pH, and arginine. We found that elevated salt concentrations, reduced pH values, or supplementation with arginine delay condensate aging (**Figure S23-S27**). Furthermore, we examined how salt concentration and pH influence reaction rates within condensates. Notably, we observed that impaired enzyme activities in aged condensates can be rescued by lower pH values and higher NaCl concentrations (**Figure S28 and S29**). Together, these results indicate that multiple factors influence condensate aging, which can be leveraged to tune enzyme reaction rates within condensates. Further studies are required to investigate the generalizability of these factors in modulating the aging of other biomolecular condensates, as different phase-forming scaffold interaction networks could yield distinct environmental sensitivities.”

Revisions to the Supporting Information:

Page S38: We have added Figure 22 as Figure S27 in the Supporting Information.

Comment 12:

In the discussion section, the statement „Despite the simplicity of this engineered system, it recapitulates key features of naturally occurring biomolecular condensates, including the selective enrichment of specific biomolecules and time-dependent changes in biophysical properties.“ is very speculative. Why is the system relevant for naturally occurring condensates.

Response 12:

Naturally occurring biomolecular condensate possesses the ability to selectively concentrate proteins. The protein constituents within these condensates are classified as either scaffolds—proteins that drive condensate formation—or clients, which preferentially partition into condensates via interactions with scaffolds (Banani et al., *Cell*, 166:651-663, 2016; Banani et al., *Nat. Rev. Mol. Cell Biol.*, 18:285-298, 2017). Previous studies have revealed that the constituents of biomolecular condensates can be concentrated by several-folds relative to their surrounding milieu (Lyon et al., *Nat. Rev. Mol. Cell Biol.*, 22:215-235, 2021). In the context of our study, intrinsically disordered FIB1 fusion proteins function as phase-forming scaffolds mediating the formation of biomolecular condensates, while enzymes are enriched within the condensates through RIAD-RIDD interactions, closely resembling the selective enrichment observed in naturally occurring condensates.

Time-dependent properties represent another key feature of naturally occurring biomolecular condensates. Particularly, many IDRs that initially form liquid-like condensate transition to solid-like state over time, a phenomenon known as condensate aging or maturation. This metastability has been observed both in vitro and in living cells (Patel et al., *Cell*, 162:1066-1077, 2015; Woodruff et al., *Cell*, 169:1066-1077, 2017; Franzmann et al., *Science*, 359:eaa05654, 2018), leading researchers to propose that condensate aging may be an intrinsic characteristic of naturally occurring biomolecular condensate (Hilditch et al. *J. Am. Chem. Soc.*, 146:10240-10245, 2024; Boeynaems et al., *Trends Cell Biol.*, 28:420-435, 2018). For example, Patel et al. found that amyotrophic lateral sclerosis (ALS)-associated protein FUS forms liquid-like condensates in cells, which subsequently convert into an aberrant aggregated state over time (Patel et al., *Cell*, 162:1066-1077, 2015). Condensate aging is characterized by restricted molecular diffusion, resistance to fusion, and loss of spherical morphology, which are also observed in our synthetic condensates, as evidenced by FRAP, TEM, and tomography analysis. Therefore, our synthetic system captures a key aspect of naturally occurring condensates.

Revisions to the manuscript:

Page 21 (line 17): For clarity, we have revised the manuscript as follows: “Despite the simplicity of this engineered system, the synthetic condensates recapitulate key features of naturally occurring biomolecular condensates. Their formation is driven by weak interactions among IDR (here, FIB1-GFP-RIAD), and enzymes are recruited and enriched through interaction with the scaffolds ^{6, 63}. Additionally, the synthetic condensates exhibited restricted molecular diffusion, resistance to fusion, and loss of spherical morphology over time, akin to the aging or maturation of naturally occurring condensates ^{25, 64, 65}.”

Page 21 (line 21): The following references have been added as Ref. 6 and Ref. 63:

1. Lyon et al. A framework for understanding the functions of biomolecular condensates across scales. *Nat. Rev. Mol. Cell Biol.*, **22**:215-235 (2021).
2. Salman F Banani et al. Compositional control of phase-separated cellular bodies. *Cell*, **166**:651-633 (2016).

Page 21 (line 23): The following references have been added as Ref. 25, Ref. 64 and Ref. 65:

1. Patel et al. A liquid-to-solid phase transition of the ALS protein FUS accelerated by disease mutation. *Cell*, **162**:1066-1077 (2015).
2. Woodruff et al. The centrosome is a selective condensate that nucleates microtubules by concentrating tubulin. *Cell*, **169**:1066-1077 (2017).
3. Franzmann et al. Phase separation of a yeast prion protein promotes cellular fitness. *Science*, **359**: eaa05654 (2018).

Responses to the Reviewers' Comments

We were pleased to receive constructive comments on the quality and significance of our work. In the revised version of the manuscript, we addressed all of the reviewers' comments point-by-point. These minor revisions have enhanced the clarity and precision of the manuscript.

Comments to all Reviewers:

We thank all of the reviewers for their insightful comments on our manuscript. In the following section, we describe the specific changes/ corrections made to the manuscript based on reviewers' comments. Reviewer comments are listed first in *italic*, while our responses follow in plain text (color in blue), and changes/additions to the manuscript text are shown in as "quoted text" (color in red).

Response to Reviewer 1

We sincerely appreciate the reviewers' recognition of the quality and significance of our work, as well as the encouraging remark, "*I recommend this manuscript for publication.*" We also thank the reviewer for the valuable minor comments, which have further improved the manuscript.

Response to comments:

Comment 1:

The added background on the passive rheology method is valuable, but the sentences beginning on line 322 reads too informally for a scientific paper. Please revise the paragraph to a more formal academic style.

Response 1:

We agree with the reviewer. The explanation about the G' and G'' turns out to be unnecessary as they are very basic mechanical properties. We thus have removed the related explanation.

Revisions to the manuscript:

Line 310 (right): We have deleted the following statement: "The storage modulus G' characterizes the elastic response of the droplets, showing how much energy the material can store and bounce back. It's about the material's stiffness, acting like a solid. The loss modulus G'' characterizes the viscous component of the response, showing how much energy the material loses as heat. It's about the material's flow, acting like a liquid."

Comment 2:

The previous comment on surface tension was to discuss surface tension in the context of compartmentalization (around line 700). The sentence added at line 182, while interesting, feels out of place in my opinion.

Response 2:

Thanks for this comment. We have removed the sentence added at line 182 (now at line 174).

Revisions to the manuscript:

Line 174: The following sentence has been deleted from the manuscript: “It is noteworthy that the size distribution and the maintenance of sub-phases of the condensates may be regulated in the presence of surface tension effects.”

Response to Reviewer 2

We sincerely appreciate the reviewers' incisive insights on the detailed aspects of our work. The constructive suggestions have substantially enhanced the accuracy and precision of the manuscript.

Response to comments:

Comment 1:

I would just like to kindly point out typos in the following sentence: "The discrepancy in the diamtere of biomolecular condensates at h0 between TEM and confocal imaging is likely attributed to uranyl acetate staining during TEM preparation (Figure S11)." Additionally, Figure 3f was not updated to Ångström in the revised version of the manuscript.

Response 1:

We thank the reviewer for bringing this out. We have corrected the typo in the manuscript. Additionally, Figure 3f has been updated with Ångström (Å) units as required (**Figure 1**).

Figure 1. (also Figure 3f in the manuscript). TEM images of catalytic condensates at different stages. Particles with increased electron density are observed over time, indicating the formation of protein aggregates (top). Higher magnification images reveal a disorder-to-order transition, with an increase in crystallized area exhibiting an approximately 3.5 Å lattice spacing characteristic of β -sheet structures (bottom). Protein aggregates are indicated by red arrowheads. The white dashed lines outline the edge of the condensates. Scale bars, 200 nm (top) and 0.5 nm (bottom).

Revisions to the manuscript:

Line 355: The sentence has been corrected to read: “The discrepancy in the diameter of biomolecular condensates at h0 between TEM and confocal imaging is likely attributed to uranyl acetate staining during TEM preparation (Figure S11).”

Line 387: We have added Figure 1 as Figure 3f.

Comment 2:

I do not find the authors' response to my comment 8 acceptable. The statement “Yellow dots only represent the non-condensate, ribosome-containing area, and do not correlate to any real cellular structures.” is problematic. Markers that do not correspond to any actual cellular structures lack scientific meaning and risk misleading the reader. To avoid confusion, the yellow dots should be removed from the figure.

Response 2:

We thank the reviewer for the comments. The yellow dots have been removed from the figures (**Figures 2** and **Figures 3**).

Figure 2. (also Figure 5c in the manuscript). Three-dimensional (3D) reconstructions of complex cellular structures. 3D models generated from tomographic reconstructions (top). Colored boxes indicate condensates at various stages and magnifications (bottom): (i), early-stage condensates ($t_i = 2$ hours); (ii-iv), middle-stage condensates ($t_i = 8$ hours) with protein aggregates; (v and vi), late-stage condensates ($t_i = 14$ hours) showing protein aggregates and cytosolic infiltration. Cell membranes are highlighted in red, protein aggregates in white (middle-stage condensates) and blue (late-stage condensates). The colored surfaces around the condensates represent pseudo-barriers between condensates and cytoplasm.

Figure 3. (also Supplementary Figure 18b in the Supplementary Information). Three-dimensional (3D) models generated from tomographic reconstructions. Cell membranes are highlighted in red and protein aggregates in white (middle-stage condensates) and blue (late-stage condensates). The colored surfaces around the condensates represent pseudo-barriers between condensates and cytoplasm.

Revisions to the Manuscript:

Line 526: We have revised the manuscript as follows: “as evidenced by the infiltration of cytoplasm into the synthetic condensates (Fig. 5c, Supplementary Fig. 18 and Supplementary Movie 3).”

Line 480: We have replaced Figure 5c with Figure 2.

Revisions to the Supplementary Information:

Page S18: We have replaced Supplementary Figure 18b with Figure 3.